# Block-Coordinate Methods and Restarting for Solving Extensive-Form Games*

**Darshan Chakrabarti**
IEOR Department
Columbia University
New York, NY 10025
dc3595@columbia.edu

**Jelena Diakonikolas**
Department of Computer Sciences
University of Wisconsin-Madison
Madison, WI 53706
jelena@cs.wisc.edu

**Christian Kroer**
IEOR Department
Columbia University
New York, NY 10025
ck2945@columbia.edu

## Abstract

Coordinate descent methods are popular in machine learning and optimization for their simple sparse updates and excellent practical performance. In the context of large-scale sequential game solving, these same properties would be attractive, but until now no such methods were known, because the strategy spaces do not satisfy the typical separable block structure exploited by such methods. We present the first cyclic coordinate-descent-like method for the polytope of sequence-form strategies, which form the strategy spaces for the players in an extensive-form game (EFG). Our method exploits the recursive structure of the proximal update induced by what are known as dilated regularizers, in order to allow for a pseudo block-wise update. We show that our method enjoys a $O(1/T)$ convergence rate to a two-player zero-sum Nash equilibrium, while avoiding the worst-case polynomial scaling with the number of blocks common to cyclic methods. We empirically show that our algorithm usually performs better than other state-of-the-art first-order methods (i.e., mirror prox), and occasionally can even beat CFR$^+$, a state-of-the-art algorithm for numerical equilibrium computation in zero-sum EFGs. We then introduce a *restarting* heuristic for EFG solving. We show empirically that restarting can lead to speedups, sometimes huge, both for our cyclic method, as well as for existing methods such as mirror prox and predictive CFR$^+$.

## 1 Introduction

Extensive-form games (EFGs) are a broad class of game-theoretic models which are played on a tree. They can compactly model both simultaneous and sequential moves, private and/or imperfect information, and stochasticity. Equilibrium computation for a two-player zero-sum EFG can be formulated as the following bilinear saddle-point problem (BSPP)

$$\min_{\mathbf{x} \in \mathcal{X}} \max_{\mathbf{y} \in \mathcal{Y}} \langle \mathbf{M}\mathbf{x}, \mathbf{y} \rangle . \tag{PD}$$

Here, the set of strategies $\mathcal{X}, \mathcal{Y}$ for the $\mathbf{x}$ and $\mathbf{y}$ players are convex polytopes known as *sequence-form polytopes* [44]. The (PD) formulation lends itself to first-order methods (FOMs) [14, 25], linear

---

*Authors are ordered alphabetically.

37th Conference on Neural Information Processing Systems (NeurIPS 2023).

programming [44], and online learning-based approaches [6, 8, 15, 17, 42, 49], since the feasible sets are convex and compact polytopes, and the objective is bilinear.

A common approach for solving BSPPs is by using first-order methods, where local gradient information is used to iteratively improve the solution in order to converge to an equilibrium asymptotically. In the game-solving context, such methods rely on two oracles: a *first-order oracle* that returns a (sub)gradient at the current pair of strategies, and a pair of *prox oracles* for the strategy spaces $\mathcal{X}, \mathcal{Y}$, which allow one to perform a generalized form of projected gradient descent steps on $\mathcal{X}, \mathcal{Y}$. These prox oracles are usually constructed through the choice of an appropriate *regularizer*. For EFGs, it is standard to focus on regularizers for which the prox oracle can be computed in linear time with respect to the size of the polytope, which is only known to be achievable through what is known as *dilated* regularizers [23]. Most first-order methods for EFGs require full-tree traversals for the first-order oracle, and full traversals of the decision sets for the prox computation, before making a strategy update for each player. For large EFGs these full traversals, especially for the first-order oracle, can be very expensive, and it may be desirable to make strategy updates before a full traversal has been performed, in order to more rapidly incorporate partial first-order information.

In other settings, one commonly used approach for solving large-scale problems is through *coordinate methods* (CMs) [34, 46]. These methods involve computing the gradient for a restricted set of coordinates at each iteration of the algorithm, and using these partial gradients to construct descent directions. The convergence rate of these methods typically is able to match the rate of full gradient methods. However, in some cases they may exhibit worse runtime due to constants introduced by the method. In spite of this, they often serve practical benefits of being more time and space efficient, and enabling distributed computation [2, 3, 11, 19, 22, 28, 30, 32, 34, 47, 48].

Generally, coordinate descent methods assume that the problem is separable, i.e., there exists a partition of the coordinates into blocks so that the feasible set can be decomposed as a Cartesian product of feasible sets, one for each block. This assumption is crucial, as it allows the methods to perform block-wise updates without worrying about feasibility, and it simplifies the convergence analysis. Extending CDMs to EFGs is non-trivial because the constraints of the sequence-form polytope do not possess this separable structure; instead the strategy space is such that the decision at a given decision point affects all variables that occur after that decision. We are only aware of a couple examples in the literature where separability is not assumed [1, 10], but those methods require strong assumptions which are not applicable in EFG settings.

**Contributions.** We propose the *Extrapolated Cyclic Primal-Dual Algorithm (ECyclicPDA)*. Our algorithm is the first cyclic coordinate method for the polytope of sequence-form strategies. It achieves a $O(1/T)$ convergence rate to a two-player zero-sum Nash equilibrium, with no dependence on the number of blocks; this, is in contrast with the worst-case polynomial dependence on the number of blocks that commonly appears in convergence rate guarantees for cyclic methods. Our method crucially leverages the recursive structure of the prox updates induced by dilated regularizers. In contrast to true cyclic (block) coordinate descent methods, the intermediate iterates generated during one iteration of ECyclicPDA are not feasible because of the non-separable nature of the constraints of sequence-form polytopes. Due to this infeasibility we refer to our updates as being pseudo-block updates. The only information that is fully determined after one pseudo-block update, is the behavioral strategy for all sequences at decision points in the block that was just considered. The behavioral strategy is converted back to sequence-form at the end of a full iteration of our algorithm.

At a very high level, our algorithm is inspired by the CODER algorithm due to Song and Diakonikolas [39]. However, there are several important differences due to the specific structure of the bilinear problem (PD) that we solve. First of all, the CODER algorithm is not directly applicable to our setting, as the feasible set (treeplex) that appears in our problem formulation is not separable. Additionally, CODER only considers Euclidean setups with quadratic regularizers, whereas our work considers more general normed settings; in particular, the $\ell_1$ setup is of primary interest for our problem setup, since it yields a much better dependence on the game size.

These two issues regarding the non-separability of the feasible set and the more general normed spaces and regularizers are handled in our work by (i) considering dilated regularizers, which allow for blockwise (up to scaling) updates in a bottom-up fashion, respecting the treeplex ordering; and (ii) introducing different extrapolation steps (see Lines 10 and 13 in Algorithm 1) that are unique to our work and specific to the bilinear EFG problem formulation. Additionally, our special problem structure and the choice of the extrapolation sequences $\widetilde{\mathbf{x}}_k$ and $\widetilde{\mathbf{y}}_k$ allows us to remove

any nonstandard Lipschitz assumptions used in Song and Diakonikolas [39]. Notably, unlike Song and Diakonikolas [39] and essentially all the work on *cyclic* methods we are aware of, which pay polynomially for the number of blocks in the convergence bound, our convergence bound in the $\ell_1$ setting is never worse than the optimal bound of full vector-update methods such as mirror prox [33] and dual extrapolation [35], which we consider a major contribution of our work.

Numerically, we demonstrate that our algorithm performs better than mirror prox (MP), and can be competitive with CFR$^+$ and its variants on certain domains. We also propose the use of *adaptive restarting* as a general heuristic tool for EFG solving: whenever an EFG solver constructs a solution with duality gap at most a constant fraction of its initial value since the last restart, we restart it and initialize the new run with the output solution at restart. Restarting is theoretically supported by the fact that BSPPs possess the *sharpness property* [5, 18, 21, 43], and restarting combined with certain *Euclidean-based* FOMs leads to a linear convergence rate under sharpness [5, 21]. We show that with restarting, it is possible for our ECyclicPDA methods to outperform CFR$^+$ on some games; this is the first time that a FOM has been observed to outperform CFR$^+$ on non-trivial EFGs. Somewhat surprisingly, we then show that for some games, restarting can drastically speed up CFR$^+$ as well. In particular, we find that on one game, CFR$^+$ with restarting exhibits a linear convergence rate, and so does a recent predictive variant of CFR$^+$ [15], on the same game and on an additional one.

**Related Work.** CMs have been widely studied in the past decade and a half [1–3, 7, 10, 11, 19, 22, 28, 30, 32, 34, 37–39, 46, 48]. CMs can be grouped into three broad classes [38]: greedy methods, which greedily select coordinates that will lead to the largest progress; randomized methods, which select (blocks of) coordinates according to a probability distribution over the blocks; and cyclic methods, which make updates in cyclic orders. Because greedy methods typically require full gradient evaluation (to make the greedy selection), the focus in the literature has primarily been on randomized (RCMs) and cyclic (CCMs) variants. RCMs require separability of the problem's constraints so we focus on CCMs. However, establishing convergence arguments for CCMs through connections with convergence arguments for full gradient methods is difficult. Some guarantees have been provided in the literature, either making restrictive assumptions [37] or by treating the cyclical coordinate gradient as an approximation of a full gradient [7], and thus incurring a linear dependence on the number of blocks in the convergence guarantee. Song and Diakonikolas [39] were the first make an improvement on reducing the dependence on the number of blocks by using a novel extrapolation strategy and introducing new block Lipschitz assumptions. That paper was the main inspiration for our work, but inapplicable to our setting, thus necessitating new technical ideas, as already discussed. While primal-dual coordinate methods for bilinear saddle-point problems have been explored in Carmon et al. [9], their techniques are not clearly extendable to our problem. The $\ell_1 - \ell_1$ setup they consider is the one which is relevant to the two-player zero-sum game setting we study, but their assumption in that case is that the feasible set for each of the players is a probability simplex, which is a much simpler feasible set than the treeplex considered in our work. It is unclear how to generalize their result to our setting, as their results depend on the simplex structure.

There has also been significant work on FOMs for two-player zero-sum EFG solving. Because this is a BSPP, off-the-shelf FOMs for BSPPs can be applied, with the caveat that proximal oracles are required. The most popular proximal oracles have been based on dilated regularizers [23], which lead to a proximal update that can be performed with a single pass over the decision space, and strong theoretical dependence on game constants [13, 14, 23, 25]. A second popular approach is the counterfactual regret minimization (CFR) framework, which decomposes regret minimization on the EFG decision sets into local simplex-based regret minimization [49]. In theory, CFR-based results have mostly led to an inferior $T^{-1/2}$ rate of convergence, but in practice the CFR framework instantiated with *regret matching*$^+$ (RM$^+$) [41] or *predictive* RM$^+$ (PRM$^+$) [15] is the fastest approach for essentially every EFG setting. The most competitive FOM-based approaches for practical performance are based on dilated regularizers [14, 24], but these have not been able to beat CFR$^+$ on EFG settings; we show for the first time that it *is* possible to beat CFR$^+$ through a combination of block-coordinate updates and restarting, at least on some games. An extended discussion of FOM and CFR approaches to EFG solving is given in Appendix A.

## 2   Notation and Preliminaries

In this section, we provide the necessary background and notation subsequently used to describe and analyze our algorithm. As discussed in the introduction, our focus is on bilinear problems (PD).

## 2.1 Notation and Optimization Background

We use bold lowercase letters to denote vectors and bold uppercase letters to denote matrices. We use $\|\cdot\|$ to denote an arbitrary $\ell_p$ norm for $p \geq 1$ applied to a vector in either $\mathbb{R}^m$ or $\mathbb{R}^n$, depending on the context. The norm dual to $\|\cdot\|$ is denoted by $\|\cdot\|_*$ and defined in the standard way as $\|\mathbf{z}\|_* = \sup_{\mathbf{x} \neq \mathbf{0}} \frac{\langle \mathbf{z}, \mathbf{x} \rangle}{\|\mathbf{x}\|}$, where $\langle \mathbf{z}, \mathbf{x} \rangle$ denotes the standard inner product. In particular, for $\|\cdot\| = \|\cdot\|_p$, where $p \geq 1$, we have $\|\cdot\|_* = \|\cdot\|_{p^*}$, where $\frac{1}{p} + \frac{1}{p^*} = 1$. We further use $\|\cdot\|_*$ to denote the induced matrix norm defined by $\|\mathbf{M}\|_* = \sup_{\mathbf{x} \neq \mathbf{0}} \frac{\|\mathbf{M}\mathbf{x}\|_*}{\|\mathbf{x}\|}$. In particular, for the Euclidean norm $\|\cdot\| = \|\cdot\|_2$, the dual norm $\|\cdot\|_* = \|\cdot\|_2$ is also the Euclidean norm, and $\|\mathbf{M}\|_* = \|\mathbf{M}\|_2$ is the matrix operator norm. For the $\ell_1$ norm $\|\cdot\| = \|\cdot\|_1$, the dual norm is the $\ell_\infty$-norm, $\|\cdot\|_* = \|\cdot\|_\infty$, while the matrix norm is $\|\mathbf{M}\|_* = \|\mathbf{M}\|_{\infty \to 1} = \sup_{\mathbf{x} \neq \mathbf{0}} \frac{\|\mathbf{M}\mathbf{x}\|_\infty}{\|\mathbf{x}\|_1} = \max_{i,j} |M_{ij}|$. We use $\Delta^n = \{\mathbf{x} \in \mathbb{R}^n : \mathbf{x} \geq \mathbf{0}, \langle \mathbf{1}, \mathbf{x} \rangle = 1\}$ to denote the probability simplex in $n$ dimensions.

**Primal-dual Gap.** Given $\mathbf{x} \in \mathbb{R}^d$, the *primal value* of the problem (PD) is $\max_{\mathbf{v} \in \mathcal{X}} \langle \mathbf{M}\mathbf{x}, \mathbf{v} \rangle$. Similarly, the *dual value* of (PD) is defined by $\min_{\mathbf{u} \in \mathcal{Y}} \langle \mathbf{M}\mathbf{u}, \mathbf{y} \rangle$. Given a primal-dual pair $(\mathbf{x}, \mathbf{y}) \in \mathcal{X} \times \mathcal{Y}$, the primal-dual gap (or saddle-point gap) is defined by

$$\text{Gap}(\mathbf{x}, \mathbf{y}) = \max_{\mathbf{v} \in \mathcal{X}} \langle \mathbf{M}\mathbf{x}, \mathbf{v} \rangle - \min_{\mathbf{u} \in \mathcal{Y}} \langle \mathbf{M}\mathbf{u}, \mathbf{y} \rangle = \max_{(\mathbf{u}, \mathbf{v}) \in \mathcal{X} \times \mathcal{Y}} \text{Gap}^{\mathbf{u}, \mathbf{v}}(\mathbf{x}, \mathbf{y}),$$

where we define $\text{Gap}^{\mathbf{u}, \mathbf{v}}(\mathbf{x}, \mathbf{y}) = \langle \mathbf{M}\mathbf{x}, \mathbf{v} \rangle - \langle \mathbf{M}\mathbf{u}, \mathbf{y} \rangle$. For our analysis, it is useful to work with the relaxed gap $\text{Gap}^{\mathbf{u}, \mathbf{v}}(\mathbf{x}, \mathbf{y})$ for some arbitrary but fixed $\mathbf{u} \in \mathcal{X}, \mathbf{v} \in \mathcal{Y}$, and then draw conclusions about a candidate solution by making concrete choices of $\mathbf{u}, \mathbf{v}$.

**Definitions and Facts from Convex Analysis.** In this paper, we primarily work with convex functions $f : \mathbb{R}^n \to \mathbb{R} \cup \{\pm\infty\}$ that are differentiable on the interior of their domain. We say that $f$ is $c_f$-strongly convex w.r.t. a norm $\|\cdot\|$ if $\forall \mathbf{y} \in \mathbb{R}^n, \forall \mathbf{x} \in \text{int dom} f$,

$$f(\mathbf{y}) \geq f(\mathbf{x}) + \langle \nabla f(\mathbf{x}), \mathbf{y} - \mathbf{x} \rangle + \frac{c_f}{2} \|\mathbf{y} - \mathbf{x}\|^2.$$

We will also need convex conjugates and Bregman divergences. Given an extended real valued function $f : \mathbb{R}^n \to \mathbb{R} \cup \{\pm\infty\}$, its convex conjugate is defined by $f^*(\mathbf{z}) = \sup_{\mathbf{z} \in \mathbb{R}^n} \{\langle \mathbf{z}, \mathbf{x} \rangle - f(\mathbf{x})\}$. Let $f : \mathbb{R}^n \to \mathbb{R} \cup \{\pm\infty\}$ be a function that is differentiable on the interior of its domain. Given $\mathbf{y} \in \mathbb{R}^n$ and $\mathbf{x} \in \text{int dom} f$, the Bregman divergence $D_f(\mathbf{y}, \mathbf{x})$ is defined by $D_f(\mathbf{y}, \mathbf{x}) = f(\mathbf{y}) - f(\mathbf{x}) - \langle \nabla f(\mathbf{x}), \mathbf{y} - \mathbf{x} \rangle$. If the function $f$ is $c_f$-strongly convex, then $D_f(\mathbf{y}, \mathbf{x}) \geq \frac{c_f}{2} \|\mathbf{y} - \mathbf{x}\|^2$.

## 2.2 Extensive-Form Games: Background and Additional Notation

Extensive form games are represented by game trees. Each node $v$ in the game tree belongs to exactly one player $i \in \{1, \ldots, n\} \cup \{c\}$ whose turn it is to move. Player c is a special player called the chance player; it is used to denote random events that happen in the game, such as drawing a card from a deck or tossing a coin. At terminal nodes of the game, players are assigned payoffs. We focus on two-player zero-sum games, where $n = 2$ and payoffs sum to zero. Private information is modeled using information sets (infosets): a player cannot distinguish between nodes in the same infoset, so the set of actions available to them must be the same at each node in the infoset.

**Treeplexes.** The decision problem for a player in a perfect recall EFG can be described as follows. There exists a set of decision points $\mathcal{J}$, and at each decision point $j$ the player has a set of actions $A_j$ with $|A_j| = n_j$ actions in total. These decision points coincide with infosets in the EFG. Without loss of generality, we let there be a single root decision point, representing the first decision the player makes in the game. The choice to play an action $a \in A_j$ for a decision point $j \in \mathcal{J}$ is represented using a sequence $(j, a)$, and after playing this sequence, the set of possible next decision points is denoted by $\mathcal{C}_{j,a}$ (which may be empty in case the game terminates). The set of decisions form a tree, meaning that $\mathcal{C}_{j,a} \cap \mathcal{C}_{j',a'} = \emptyset$ unless $j = j'$ and $a = a'$; this is known as *perfect recall*. The last sequence (necessarily unique) encountered on the path from the root to decision point $j$ is denoted by $p_j$. We define $\downarrow j$ as the set consisting of all decision points that can be reached from $j$. An example of the use of this notation for a player in Kuhn poker [26] can be found in Appendix B.

The set of strategies for a player can be characterized using the *sequence-form*, where the value of the decision variable assigned to playing the sequence $(j, a)$ is the product of the decision variable assigned to playing the parent sequence $p_j$ and the probability of playing action $a$ when at $j$ [44].

The set of all sequence-form strategies of a player form a polytope known as the sequence-form polytope. Sequence-form polytopes fall into a class of polytopes known as treeplexes [23], which can be characterized inductively using convex hull and Cartesian product operations:

**Definition 2.1** (Treeplex). A treeplex $\mathcal{X}$ for a player can be characterized recursively as follows, where $r$ is the the root decision point for a player.

$$\mathcal{X}_{j,a} = \prod_{j' \in \mathcal{C}_{j,a}} \mathcal{X}_{\downarrow j'},$$

$$\mathcal{X}_{\downarrow j} = \{(\lambda_1, \ldots, \lambda_{|A_j|}, \lambda_1 \mathbf{x}_1, \ldots, \lambda_{|A_j|} \mathbf{x}_{|A_j|} : (\lambda_1, \ldots, \lambda_{|A_j|}) \in \Delta^{|A_j|}, \mathbf{x}_a \in \mathcal{X}_{j,a}\},$$

$$\mathcal{X} = \{1\} \times \mathcal{X}_{\downarrow r}.$$

This formulation allows the expected loss of a player to be formulated as a bilinear function $\langle \mathbf{M}\mathbf{x}, \mathbf{y} \rangle$ of players' strategies $\mathbf{x}, \mathbf{y}$. This gives rise to the BSPP in Equation (PD), and the set of saddle points of that BSPP are exactly the set of Nash equilibria of the EFG. The *payoff matrix* $\mathbf{M}$ is a sparse matrix, whose nonzeroes correspond to the set of leaf nodes of the game tree.

**Indexing Notation.** A sequence-form strategy of a player can be written as a vector $\mathbf{v}$, with an entry for each sequence $(j, a)$. We use $\mathbf{v}^j$ to denote the subset of size $|A_j|$ of entries of $\mathbf{v}$ that correspond to sequences $(j, a)$ formed by taking actions $a \in A_j$ and let $\mathbf{v}^{\downarrow j}$ denote the subset of entries of $\mathbf{v}$ that are indexed by sequences that occur in the subtreeplex rooted at $j$. Additionally, we use $v^{p_j}$ to denote the (scalar) value of the parent sequence of decision point $j$. By convention, for the root decision point $j$, we let $v^{p_j} = 1$. Observe that for any $j \in \mathcal{J}$, $\mathbf{v}^j / v^{p_j}$ is in the probability simplex.

Given a treeplex $\mathcal{Z}$ we denote by $\mathcal{J}_{\mathcal{Z}}$ the set of infosets for this treeplex. We say that a partition of $\mathcal{J}_{\mathcal{Z}}$ into $k \leq |\mathcal{J}_{\mathcal{Z}}|$ sets $\mathcal{J}_{\mathcal{Z}}^{(1)}, \ldots, \mathcal{J}_{\mathcal{Z}}^{(k)}$ respects the treeplex ordering if for any two sets $\mathcal{J}_{\mathcal{Z}}^{(i)}$, $\mathcal{J}_{\mathcal{Z}}^{(i')}$ with $i < i'$ and any two infosets $j \in \mathcal{J}_{\mathcal{Z}}^{(i)}$, $j' \in \mathcal{J}_{\mathcal{Z}}^{(i')}$, $j$ does not intersect the path from $j'$ to the root decision point. The set of infosets for the player $\mathbf{x}$ is denoted by $\mathcal{J}_{\mathcal{X}}$, while the set of infosets for player $\mathbf{y}$ is denoted by $\mathcal{J}_{\mathcal{Y}}$. We assume that $\mathcal{J}_{\mathcal{X}}$ and $\mathcal{J}_{\mathcal{Y}}$ are partitioned into $s$ nonempty sets $\mathcal{J}_{\mathcal{X}}^{(1)}, \mathcal{J}_{\mathcal{X}}^{(2)}, \ldots, \mathcal{J}_{\mathcal{X}}^{(s)}$ and $\mathcal{J}_{\mathcal{Y}}^{(1)}, \mathcal{J}_{\mathcal{Y}}^{(2)}, \ldots, \mathcal{J}_{\mathcal{Y}}^{(s)}$, where $s \leq \min\{|\mathcal{J}_{\mathcal{X}}|, |\mathcal{J}_{\mathcal{Y}}|\}$ and the ordering of the sets in the two partitions respect the treeplex ordering of $\mathcal{X}, \mathcal{Y}$, respectively.

Given a pair $(t, t')$, we use $\mathbf{M}_{t,t'}$ to denote the full-dimensional $(m \times n)$ matrix obtained from the matrix $\mathbf{M}$ by keeping all entries indexed by $\mathcal{J}_{\mathcal{X}}^{(t)}$ and $\mathcal{J}_{\mathcal{Y}}^{(t')}$, and zeroing out the rest. When in place of $t$ or $t'$ we use ":", it corresponds to keeping as non-zeros all rows (for the first index) or all columns (for the second index). In particular, $\mathbf{M}_{t,:}$ is the matrix that keeps all rows of $\mathbf{M}$ indexed by $\mathcal{J}_{\mathcal{X}}^{(t)}$ intact and zeros out the rest. Further, notation $\mathbf{M}_{t',t:s}$ is used to indicate that we select rows indexed by $\mathcal{J}_{\mathcal{X}}^{(t')}$ and all columns of $\mathbf{M}$ indexed by $\mathcal{J}_{\mathcal{Y}}^{(t)}, \mathcal{J}_{\mathcal{Y}}^{(t+1)}, \ldots, \mathcal{J}_{\mathcal{Y}}^{(s)}$, while we zero out the rest; similarly for $\mathbf{M}_{t:s,t'}$. Notation $\mathbf{M}_{t',1:t}$ is used to indicate that we select rows indexed by $\mathcal{J}_{\mathcal{X}}^{(t')}$ and all columns of $\mathbf{M}$ indexed by $\mathcal{J}_{\mathcal{Y}}^{(1)}, \mathcal{J}_{\mathcal{Y}}^{(2)}, \ldots, \mathcal{J}_{\mathcal{Y}}^{(t)}$, while we zero out the rest; similarly for $\mathbf{M}_{1:t,t'}$. Given a vector $\mathbf{x} \in \mathcal{X}$, $\mathbf{x}^{(t)}$ denotes the entries of $\mathbf{x}$ indexed by the elements of $\mathcal{J}_{\mathcal{X}}^{(t)}$; similarly, for $\mathbf{y} \in \mathcal{Y}$, $\mathbf{y}^{(t)}$ denotes the entries of $\mathbf{y}$ indexed by the elements of $\mathcal{J}_{\mathcal{Y}}^{(t)}$.

Additionally, we use $\mathbf{M}^{(t,t')}$ to denote the submatrix of $\mathbf{M}$ obtained by selecting rows indexed by $\mathcal{J}_{\mathcal{X}}^{(t)}$ and columns indexed by $\mathcal{J}_{\mathcal{Y}}^{(t')}$. $\mathbf{M}^{(t,t')}$ is $(p \times q)$-dimensional, for $p = \sum_{j \in \mathcal{J}_{\mathcal{X}}^{(t)}} |A_j|$ and $q = \sum_{j \in \mathcal{J}_{\mathcal{Y}}^{(t')}} |A_j|$. Notation ":" has the same meaning as in the previous paragraph.

**Dilated Regularizers.** We assume access to strongly convex functions $\phi : \mathcal{X} \to \mathbb{R}$ and $\psi : \mathcal{Y} \to \mathbb{R}$ with known strong convexity parameters $c_\phi > 0$ and $c_\psi > 0$, and that are continuously differentiable on the interiors of their respective domains. We further assume that these functions are *nice* as defined by Farina et al. [14]: their gradients and the gradients of their convex conjugates can be computed in time linear (or nearly linear) in the dimension of the treeplex.

A dilated regularizer is a framework for constructing nice regularizing functions for treeplexes. It makes use of the inductive characterization of a treeplex via Cartesian product and convex hull operations to generalize from the local simplex structure of the sequence-form polytope at a decision point to the entire sequence-form polytope. In particular, given a local "nice" regularizer $\phi^j$ for each decision point $j$, a dilated regularizer for the treeplex can be defined as $\phi(\mathbf{x}) = \sum_{j \in \mathcal{J}_{\mathcal{X}}} x^{p_j} \phi^j \left( \frac{\mathbf{x}^j}{x^{p_j}} \right)$.

The key property of these dilated regularizing functions is that the prox computations of the form $\mathbf{x}_k = \arg\min_{\mathbf{x}\in\mathcal{X}}\{\langle \mathbf{h}, \mathbf{x}\rangle + D_\phi(\mathbf{x}_k, \mathbf{x}_{k-1})\}$ decompose into bottom-up updates, where, up to a scaling factor, each set of coordinates from set $\mathcal{J}_\mathcal{X}^{(t)}$ can be computed solely based on the coordinates of $\mathbf{x}_k$ from sets $\mathcal{J}_\mathcal{X}^{(1)}, \ldots, \mathcal{J}_\mathcal{X}^{(t-1)}$ and coordinates of $\mathbf{g}$ from sets $\mathcal{J}_\mathcal{X}^{(1)}, \ldots, \mathcal{J}_\mathcal{X}^{(t)}$. Concretely, the recursive structure of the prox update is as follows (this was originally shown by [23], here we show a variation from Farina et al. [13]):

**Proposition 2.2** (Farina et al. [13]). *A prox update to compute $\mathbf{x}_k$, with gradient $\mathbf{h}$ and center $\mathbf{x}_{k-1}$ on a treeplex $\mathcal{X}$ using a Bregman divergence constructed from a dilated DGF $\phi$ can be decomposed into local prox updates at each decision point $j \in \mathcal{J}_\mathcal{X}$ as follows:*

$$\mathbf{x}_k^j = \mathbf{x}_k^{p_j} \cdot \underset{\mathbf{b}^j \in \Delta^{n_j}}{\arg\min}\left\{ \left\langle \mathbf{h}^j + \hat{\mathbf{h}}^j, \mathbf{b}^j\right\rangle + D_{\phi^j}\left(\mathbf{b}^j, \frac{\mathbf{x}_{k-1}^j}{\mathbf{x}_{k-1}^{p_j}}\right)\right\},$$

$$\hat{h}^{(j,a)} = \sum_{j' \in \mathcal{C}_{j,a}}\left[ \phi^{\downarrow j'^*}\left(-\mathbf{h}^{\downarrow j} + \nabla\phi^{\downarrow j'}\left(x_{k-1}^{\downarrow j'}\right)\right) - \phi^{j'}\left(\frac{\mathbf{x}_{k-1}^{j'}}{x_{k-1}^{(j,a)}}\right) + \left\langle \nabla\phi^{j'}\left(\frac{\mathbf{x}_{k-1}^{j'}}{x_{k-1}^{(j,a)}}\right), \frac{\mathbf{x}_{k-1}^{j'}}{x_{k-1}^{(j,a)}}\right\rangle\right].$$

## 3 Extrapolated Cyclic Algorithm

Our extrapolated cyclic primal-dual algorithm is summarized in Algorithm 1. As discussed in Section 2, under the block partition and ordering that respects the treeplex ordering, the updates for $\mathbf{x}_k^{(t)}$ in Line 9 (respectively, $\mathbf{y}_k^{(t)}$ in Line 12), up to scaling by the value of their respective parent sequences, can be carried out using only the information about $\frac{\mathbf{x}_k^j}{x_k^{p_j}}$ and $\mathbf{h}_k^j$ (respectively, $\frac{\mathbf{y}_k^j}{y_k^{p_j}}$ and $\mathbf{g}_k^j$) for infosets $j$ that are "lower" on the treeplex. The specific choices of the extrapolation sequences $\widetilde{\mathbf{x}}_k$ and $\widetilde{\mathbf{y}}_k$ that only utilize the information from prior cycles and the scaled values of $\frac{\mathbf{x}_k^j}{x_k^{p_j}}$ and $\frac{\mathbf{y}_k^j}{y_k^{p_j}}$ for infosets $j$ updated up to the block $t$ updates for $\mathbf{x}_k$ and $\mathbf{y}_k$ are what crucially enables us to decompose the updates for $\mathbf{x}_k$ and $\mathbf{y}_k$ into local block updates carried out in the bottom-up manner. At the end of the cycle, once $\frac{\mathbf{x}_k^j}{x_k^{p_j}}$ and $\frac{\mathbf{y}_k^j}{y_k^{p_j}}$ has been updated for all infosets, we can carry out a top-to-bottom update to fully determine vectors $\mathbf{x}_k$ and $\mathbf{y}_k$, as summarized in the last two for loops in Algorithm 1. We present an implementation-specific version of the algorithm in Appendix D, which explicitly demonstrates that our algorithm's runtime does not have a dependence on the number of blocks used. In our analysis of the implementation-specific version of the algorithm, we argue that the per-iteration complexity of our algorithm matches that of MP. To support this analysis, we compare the empirical runtimes of our algorithm with MP and CFR$^+$ variants in Section 4.

Our convergence argument is built on the decomposition of the relaxed gap $\mathrm{Gap}^{\mathbf{u},\mathbf{v}}(\mathbf{x}_k, \mathbf{v}_k)$ for arbitrary but fixed $(\mathbf{u}, \mathbf{v}) \in \mathcal{X} \times \mathcal{Y}$ into telescoping and non-positive terms, which is common in first-order methods. The first idea that enables leveraging cyclic updates lies in replacing vectors $\mathbf{M}\mathbf{x}_k$ and $\mathbf{M}^\top \mathbf{y}_k$ by "extrapolated" vectors $\mathbf{g}_k$ and $\mathbf{h}_k$ that can be partially updated in a blockwise fashion as a cycle of the algorithm progresses, as stated in Proposition 3.1. To our knowledge, this basic idea originates in Song and Diakonikolas [39]. Unique to our work are the specific choices of $\mathbf{g}_k$ and $\mathbf{h}_k$, which leverage all the partial information known to the algorithm up to the current iteration and block update. Crucially, we leverage the treeplex structure to show that our chosen updates are sufficient to bound the error sequence $\mathcal{E}_k$ and obtain the claimed convergence bound in Theorem 3.2. Due to space constraints, the proof is deferred to Appendix C.

To simplify the exposition, we introduce the following notation:

$$\mathbf{M_x} := \sum_{t=1}^{s-1} \mathbf{M}_{t,t+1:s}, \quad \mathbf{M_y} := \mathbf{M} - \mathbf{M_x} = \sum_{t=1}^{s} \mathbf{M}_{t:s,t};$$

$$\mu_x := \|\mathbf{M_x}\|_* + \|\mathbf{M_y}\|_*, \quad \mu_y := \|\mathbf{M_x}^\top\|_* + \|\mathbf{M_y}^\top\|_*. \tag{3.1}$$

When the norm of the space is $\|\cdot\| = \|\cdot\|_1$, both $\mu_x$ and $\mu_y$ are bounded above by $2\max_{i,j}|M_{ij}|$.

The next proposition decomposes the relaxed gap into an error term and telescoping terms. The proposition is independent of the specific choices of extrapolated vectors $\mathbf{g}_k, \mathbf{h}_k$.

---

**Algorithm 1** Extrapolated Cyclic Primal-Dual EFG Solver (ECyclicPDA)

---

1: **Initialization:** $\mathbf{x}_0 \in \mathcal{X}, \mathbf{y}_0 \in \mathcal{Y}, \eta_0 = H_0 = 0, \eta = \frac{\sqrt{c_\phi c_\psi}}{\mu_x + \mu_y}, \bar{\mathbf{x}}_0 = \mathbf{x}_0, \bar{\mathbf{y}}_0 = \mathbf{y}_0, \mathbf{g}_0 = \mathbf{0}, \mathbf{h}_0 = \mathbf{0}$

2: **for** $k = 1 : K$ **do**

3:      Choose $\eta_k \le \eta$, $H_k = H_{k-1} + \eta_k$

4:      $\mathbf{g}_k = \mathbf{g}_{k-1}, \mathbf{h}_k = \mathbf{h}_{k-1}$

5:      $\widetilde{\mathbf{x}}_k = \mathbf{x}_{k-1} + \frac{\eta_{k-1}}{\eta_k}(\mathbf{x}_{k-1} - \mathbf{x}_{k-2}), \widetilde{\mathbf{y}}_k = \mathbf{y}_{k-1} + \frac{\eta_{k-1}}{\eta_k}(\mathbf{y}_{k-1} - \mathbf{y}_{k-2})$

6:      **for** $t = 1 : s$ **do**

7:          $\mathbf{h}_k^{(t)} = (\mathbf{M}^{(:,t)})^\top \widetilde{\mathbf{y}}_k$

8:          $\mathbf{x}_k^{(t)} = \left[ \arg\min_{\mathbf{x} \in \mathcal{X}} \left\{ \eta_k \langle \mathbf{x}, \mathbf{h}_k \rangle + D_\phi(\mathbf{x}, \mathbf{x}_{k-1}) \right\} \right]^{(t)}$

9:          $\widetilde{\mathbf{x}}_k^{(t)} = \left[ \frac{\mathbf{x}_k^j}{x_k^{p_j}} x_{k-1}^{p_j} + \frac{\eta_{k-1}}{\eta_k} \left( \mathbf{x}_{k-1}^j - \frac{\mathbf{x}_{k-1}^j}{x_{k-1}^{p_j}} x_{k-2}^{p_j} \right) \right]_{j \in \mathcal{J}_\mathcal{X}^{(t)}}$

10:          $\mathbf{g}_k^{(t)} = \mathbf{M}^{(t,:)} \widetilde{\mathbf{x}}_k$

11:          $\mathbf{y}_k^{(t)} = \left[ \arg\max_{\mathbf{v} \in \mathcal{Y}} \left\{ \eta_k \langle \mathbf{g}_k, \mathbf{v} \rangle - D_\psi(\mathbf{v}, \mathbf{y}_{k-1}) \right\} \right]^{(t)}$

12:          $\widetilde{\mathbf{y}}_k^{(t)} = \left[ \frac{\mathbf{y}_k^j}{y_k^{p_j}} y_{k-1}^{p_j} + \frac{\eta_{k-1}}{\eta_k} \left( \mathbf{y}_{k-1}^j - \frac{\mathbf{y}_{k-1}^j}{y_{k-1}^{p_j}} y_{k-2}^{p_j} \right) \right]_{j \in \mathcal{J}_\mathcal{Y}^{(t)}}$

13:      **for** $j \in \mathcal{J}_\mathcal{X}$ **do**

14:          $\mathbf{x}_k^j = x_k^{p_j} \cdot \left( \frac{\mathbf{x}_k^j}{x_k^{p_j}} \right)$

15:      **for** $j \in \mathcal{J}_\mathcal{Y}$ **do**

16:          $\mathbf{y}_k^j = y_k^{p_j} \cdot \left( \frac{\mathbf{y}_k^j}{y_k^{p_j}} \right)$

17:      $\bar{\mathbf{x}}_k = \frac{H_k - \eta_k}{H_k} \bar{\mathbf{x}}_{k-1} + \frac{\eta_k}{H_k} \mathbf{x}_k, \bar{\mathbf{y}}_k = \frac{H_k - \eta_k}{H_k} \bar{\mathbf{y}}_{k-1} + \frac{\eta_k}{H_k} \mathbf{y}_k$

18: **Return:** $\bar{\mathbf{x}}_K, \bar{\mathbf{y}}_K$

---

**Proposition 3.1.** *Let* $\mathbf{x}_k, \mathbf{y}_k$ *be the iterates of Algorithm 1 for* $k \ge 1$. *Then, for all* $k \ge 1$, $\mathbf{x}_k \in \mathcal{X}$, $\mathbf{y}_k \in \mathcal{Y}$, *we have*

$$\eta_k \mathrm{Gap}^{\mathbf{u},\mathbf{v}}(\mathbf{x}_k, \mathbf{y}_k) \le \mathcal{E}_k - D_\phi(\mathbf{u}, \mathbf{x}_k) + D_\phi(\mathbf{u}, \mathbf{x}_{k-1}) - D_\psi(\mathbf{v}, \mathbf{y}_k) + D_\psi(\mathbf{v}, \mathbf{y}_{k-1}),$$

*where the error sequence* $\mathcal{E}_k$ *is defined by*

$$\mathcal{E}_k := \eta_k \langle \mathbf{M}\mathbf{x}_k - \mathbf{g}_k, \mathbf{v} - \mathbf{y}_k \rangle - \eta_k \langle \mathbf{u} - \mathbf{x}_k, \mathbf{M}^\top \mathbf{y}_k - \mathbf{h}_k \rangle - D_\psi(\mathbf{y}_k, \mathbf{y}_{k-1}) - D_\phi(\mathbf{x}_k, \mathbf{x}_{k-1}).$$

To obtain our main result, we leverage the blockwise structure of the problem, the bilinear structure of the objective, and the treeplex structure of the feasible sets to control the error sequence $\mathcal{E}_k$. A key property that enables this result is that normalized entries $\mathbf{x}_k^j / x_{k-1}^{p_j}$ from the same information set belong to a probability simplex. This property is crucially used in controlling the error of the extrapolation vectors. The main result is summarized in the following theorem.

**Theorem 3.2.** *Consider the iterates* $\mathbf{x}_k, \mathbf{y}_k$ *for* $k \ge 1$ *in Algorithm 1 and the output primal-dual pair* $\bar{\mathbf{x}}_K, \bar{\mathbf{y}}_K$. *Then,* $\forall k \ge 1$,

$$\frac{\mu_x D_\phi(\mathbf{x}^*, \mathbf{x}_K) + \mu_y D_\psi(\mathbf{y}^*, \mathbf{y}_K)}{\mu_x + \mu_y} \le D_\phi(\mathbf{x}^*, \mathbf{x}_0) + D_\psi(\mathbf{y}^*, \mathbf{y}_0), \quad \text{and, further,}$$

$$\mathrm{Gap}(\bar{\mathbf{x}}_K, \bar{\mathbf{y}}_K) = \sup_{\mathbf{u} \in \mathcal{X}, \mathbf{v} \in \mathcal{Y}} \{ \langle \mathbf{M}\bar{\mathbf{x}}_K, \mathbf{v} \rangle - \langle \mathbf{M}\mathbf{u}, \bar{\mathbf{y}}_K \rangle \} \le \frac{\sup_{\mathbf{u} \in \mathcal{X}, \mathbf{v} \in \mathcal{Y}} \{ D_\phi(\mathbf{u}, \mathbf{x}_0) + D_\psi(\mathbf{v}, \mathbf{y}_0) \}}{H_K}.$$

*In the above bound, if* $\forall k \ge 1$, $\eta_k = \eta = \frac{\sqrt{c_\phi c_\psi}}{\mu_x + \mu_y}$, *then* $H_K = K\eta$. *As a consequence, for any* $\epsilon > 0$, $\mathrm{Gap}(\bar{\mathbf{x}}_K, \bar{\mathbf{y}}_K) \le \epsilon$ *after at most* $\left\lceil \frac{(\mu_x + \mu_y)(\sup_{\mathbf{u} \in \mathcal{X}, \mathbf{v} \in \mathcal{Y}} \{ D_\phi(\mathbf{u}, \mathbf{x}_0) + D_\psi(\mathbf{v}, \mathbf{y}_0) \})}{\sqrt{c_\phi c_\psi} \epsilon} \right\rceil$ *iterations.*

## 4 Experimental Evaluation and Discussion

We evaluate the performance of ECyclicPDA instantiated with three different dilated regularizers: dilated entropy [25], dilatable global entropy [14], and dilated $\ell^2$ [13]. In the case of the dilated $\ell^2$

regularizer, we use dual averaging of the "extrapolated" vectors $\mathbf{g}_k$ and $\mathbf{h}_k$ in our algorithm, since otherwise we have no guarantee that the iterates would remain in the relative interior of the domain of the dilated DGF, and the Bregman divergence may become undefined. We compare our method to MP, which is state-of-the-art among first-order methods for EFG solving. We test ECyclicPDA and MP with three different averaging schemes: uniform, linear, and quadratic averaging since Gao et al. [20] suggest that these different averaging schemes can lead to faster convergence in practice. We also compare against empirical state-of-the-art CFR$^+$ variants: CFR$^+$ [41], and the predictive CFR$^+$ variant (PCFR$^+$) [15]. We emphasize that our method achieves the same $O(\frac{1}{T})$ average-iterate convergence rate as MP, and that all the CFR+ variants have the same suboptimal $O(\frac{1}{\sqrt{T}})$ average-iterate convergence rate. We experiment on four standard benchmark games for EFG solving: Goofspiel (4 ranks), Liar's Dice, Leduc (13 ranks), and Battleship. In all experiments, we run for $10,000$ full (or equivalent) gradient computations. This corresponds to 5,000 iterations of ECyclicPDA, CFR$^+$, and PCFR$^+$, and 2,500 iterations of MP.[2] A description of all games is provided in Appendix E. Additional experimental details are provided in Appendix G.

For each instantiation of ECyclicPDA considered on a given game (choice of regularizer, averaging, and block construction strategy) the stepsize is tuned by taking power of 2 multiples of $\eta$ ($2^l \cdot \eta$ for $l \in \mathbb{N}$), where $\eta$ is the theoretical stepsize stated in Theorem 3.2, and then choosing the stepsize $\eta^*$ among these multiples of $\eta$ that has the best performance. Within the algorithm, we use a constant stepsize, letting $\eta_k = \eta_0$ for all $k$. We apply the same tuning scheme for MP stepsizes (for a given choice of regularizer and averaging). Note that this stepsize tuning is coarse, and so it is possible that better results can be achieved for ECyclicPDA and MP using finer stepsize tuning.

We test our algorithm with four different block construction strategies. The *single block* construction strategy puts every decision point in a single block, and thus it corresponds to the non-block-based version of ECyclicPDA. The *children* construction strategy iterates through the decision points of the treeplex bottom-up (by definition, this will respect the treeplex ordering), and placing each set of decision points that have parent sequences starting from the same decision point in its own block. The *postorder* construction strategy iterates through the decision points bottom-up (again, by definition, this will respect the treeplex ordering). The order is given by a postorder traversal of the treeplex, treating all decision points that have the same parent sequence as a single node (and when the node is processed, all decision points are sequentially added to the block). It greedily makes blocks as large as possible, while only creating a new block if it causes a parent decision point and child decision point to end up in the same block. The *infosets* construction strategy places each decision point in its own block. We provide further description of the block construction strategies in Appendix F.

We show the results of different block construction strategies in Figure 1. For each block construction strategy, ECyclicPDA is instantiated with the choice of regularizer and averaging that yields the fastest convergence among all choices of parameters. We can see that the different block construction strategies do not make a significant difference in Goofspiel (4 ranks) or in Leduc (13 ranks). However, we see benefits of using blocks in Liar's Dice and Battleship. In Liar's Dice, children and postorder have a clear advantage, and children outperforms the other block construction strategies in Battleship.

We show the results of comparing our algorithm against MP, CFR$^+$, and PCFR$^+$ in Figure 2. ECyclicPDA is instantiated with the choice of regularizer, averaging, and block construction strategy that yields the fastest convergence among all choices for ECyclicPDA, and MP is instantiated with the choice of regularizer and averaging that yields the fastest convergence among all choices for MP. We see that ECyclicPDA performs better than MP in all games besides Goofspiel (4 ranks), where they perform about the same. In Liar's Dice and Battleship, the games where ECyclicPDA benefits from having multiple blocks, we see competitiveness with CFR$^+$ and PCFR$^+$. In particular, in Liar's Dice, ECyclicPDA is overtaking CFR$^+$ at $10,000$ gradient computations. On Battleship, we see that both ECyclicPDA and MP outperform CFR$^+$, and that ECyclicPDA is competitive with PCFR$^+$.

**Restarting.** We now introduce restarting as a heuristic tool for speeding up EFG solving. While restarting is only known to lead to a linear convergence rate in the case of using the $\ell_2$ regularizer in certain FOMs [5, 21], we apply restarting as a heuristic across our methods based on dilated regularizers and to CFR-based methods. To the best of our knowledge, restarting schemes have not been empirically evaluated on EFG algorithms such as MP, CFR$^+$, or (obviously), our new method. We implement restarting by resetting the averaging process when the duality gap has halved since the last time the averaging process was reset. After resetting, the initial iterate is set equal to the last

---

[2]Here we count one gradient evaluation for $\mathbf{x}$ and one for $\mathbf{y}$ as two gradient evaluations total.

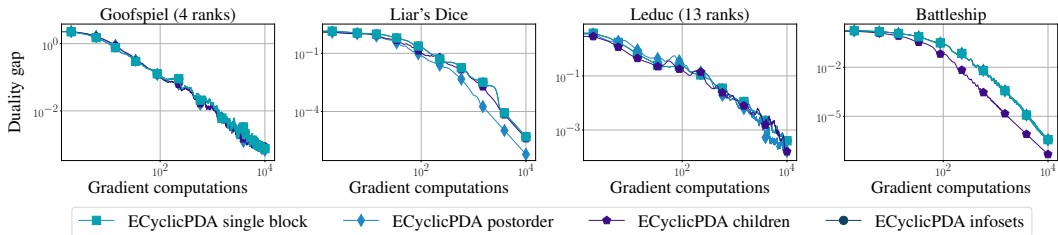

Figure 1: Duality gap as a function of the number of full (or equivalent) gradient computations for ECyclicPDA with different block construction strategies.

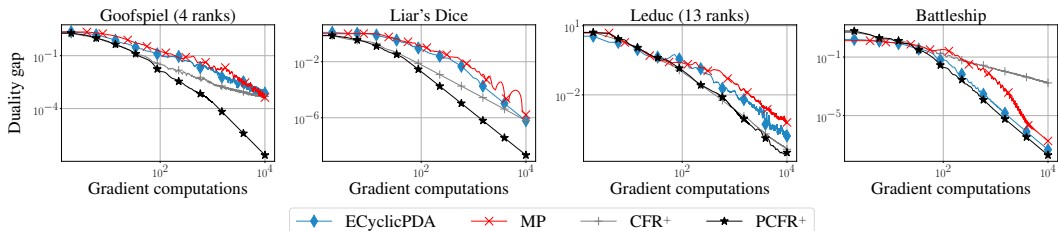

Figure 2: Duality gap as a function of the number of full (or equivalent) gradient computations for ECyclicPDA, MP, CFR$^+$, PCFR$^+$.

iterate we saw before restarting. Since the restarting heuristic is one we introduce, we distinguish restarted variants in the plots by prepending the name of the algorithm with "r". For example, when restarting is applied to CFR$^+$, the label used is rCFR$^+$.

We show the results of different block construction strategies when restarting is used on ECyclicPDA in Figure 3. As before, we take the combination of regularizer and averaging scheme that works best. Again, we can see that the different block construction strategies do not make a significant difference in Goofspiel (4 ranks) or in Leduc (13 ranks), while making a difference for Liar's Dice and Battleship. However, with restarting, the benefit of the children and postorder for Liar's Dice and Battleship is even more pronounced relative to the other block construction strategies; the gap is several orders of magnitude after $10^4$ gradient computations. Note that while children performed worse than single block for Battleship previously, with restarting, children performs much better.

Finally, we compare the performance of the restarted version of our algorithm, with restarted versions of MP, CFR$^+$, and PCFR$^+$ in Figure 4. As before, we take the combination of regularizer, averaging scheme, and block construction strategy that works best for ECyclicPDA, and the combination of regularizer and averaging scheme that works best for MP. Firstly, we note that the scale of the y-axis is different from Figure 2 for all games besides Leduc (13 ranks), because restarting tends to hit much higher levels of precision. We see that restarting provides significant benefits for PCFR$^+$ in Goofspiel (4 ranks) allowing it to converge to numerical precision, while the other algorithms do not benefit much. In Liar's Dice, restarted CFR$^+$ and PCFR$^+$ converge to numerical precision within 200 gradient computations, and restarted ECyclicPDA converges to numerical precision at $10^4$ gradient computations. Additionally, restarted MP achieves a much lower duality gap. For Battleship, ECyclicPDA, MP, and PCFR$^+$ all benefit from restarting, and restarted ECyclicPDA is competitive with restarted PCFR$^+$. Similar to the magnification in benefit of using blocks versus not using blocks when restarting in Liar's Dice and Battleship, we see that restarted ECyclicPDA achieves significantly better duality gap than MP in these games.

**Wall-clock time experiments.** In Table 1, we show the average wall-clock time per iteration of our algorithm instantiated with different block construction strategies as well as MP, CFR$^+$, and PCFR$^+$. It is clear from this table that the runtimes are pretty similar to each other, and this is without extensive optimization of our particular implementation. Clearly, our algorithm is at least as fast as MP per "full" gradient computation. Since the computational bottleneck of gradient and prox computations becomes apparent in bigger games, Battleship demonstrates the speed of our algorithm relative to MP best.

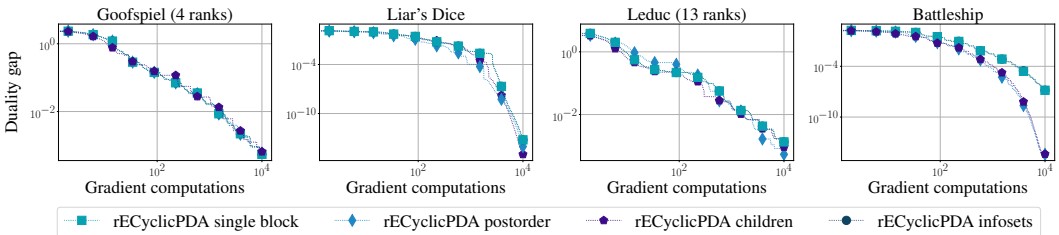

Figure 3: Duality gap as a function of the number of full (or equivalent) gradient computations for when restarting is applied to ECyclicPDA with different block construction strategies. We take the best duality gap seen so far so that the plot is monotonic.

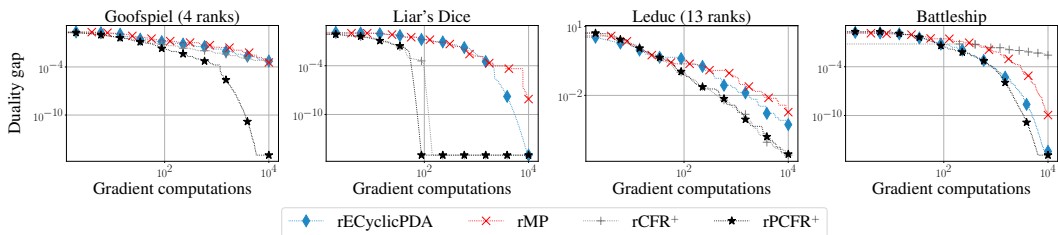

Figure 4: Duality gap as a function of the number of full (or equivalent) gradient computations for when restarting is applied to ECyclicPDA, MP, CFR$^+$, PCFR$^+$. We take the best duality gap seen so far so that the plot is monotonic.

Table 1: The average wall clock time per iteration in milliseconds of ECyclicPDA instantiated with different block construction strategies, MP, CFR$^+$, and PCFR$^+$. The duality gap is computed every 100 iterations.

| Name | Goofspiel (4 ranks) | Liar's Dice | Leduc (13 ranks) | Battleship |
|---|---|---|---|---|
| ECyclicPDA single block | 2.470 | 11.370 | 6.770 | 46.020 |
| ECyclicPDA children | 4.270 | 15.020 | 7.450 | 47.230 |
| ECyclicPDA infosets | 8.850 | 16.240 | 7.370 | 53.490 |
| ECyclicPDA postorder | 4.340 | 14.350 | 7.430 | 49.020 |
| MP | 3.720 | 19.530 | 10.360 | 83.450 |
| CFR$^+$ | 1.330 | 1.360 | 0.370 | 7.980 |
| Predictive CFR$^+$ | 1.750 | 1.920 | 0.430 | 9.410 |

**Discussion.** We develop the first cyclic block-coordinate-like method for two-player zero-sum EFGs. Our algorithm relies on the recursive nature of the prox updates for dilated regularizers, cycling through blocks that respect the partial order induced on decision points by the treeplex, and extrapolation to conduct pseudo-block updates, produce feasible iterates, and achieve $O(\frac{1}{T})$ ergodic convergence. Furthermore, the runtime of our algorithm has no dependence on the number of blocks. We present empirical evidence that our algorithm generally outperforms MP, and is the first FOM to compete with CFR$^+$ and PCFR$^+$ on non-trivial EFGs. Finally, we introduce a restarting heuristic for EFG solving, and demonstrate often huge gains in convergence rate. An open question raised by our work is understanding what makes restarting work for methods used with regularizers besides the $\ell_2$ regularizer (the only setting for which there exist linear convergence guarantees). This may be challenging because existing proofs require upper bounding the corresponding Bregman divergence (for a given non-$\ell_2$ regularizer) between iterates by the distance to optimality. This is difficult for entropy or any dilated regularizer because the initial iterate used by the algorithm after restarting may have entries arbitrarily close to zero even if they are guaranteed to not exactly be zero (as is the case for entropy). Relatedly, both our block-coordinate method and restarting have a much bigger advantage in some numerical instances (Battleship, Liar's Dice) than others (Leduc and Goofspiel); a crucial question is to understand what type of game structure drives this behavior.

## Acknowledgements

Darshan Chakrabarti was supported by National Science Foundation Graduate Research Fellowship Program under award number DGE-2036197. Jelena Diakonikolas was supported by the Office of Naval Research under award number N00014-22-1-2348. Christian Kroer was supported by the Office of Naval Research awards N00014-22-1-2530 and N00014-23-1-2374, and the National Science Foundation awards IIS-2147361 and IIS-2238960.

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

## A    Additional Related Work

There has been significant work on FOMs for two-player zero-sum EFG solving. Because this is a BSPP, off-the-shelf FOMs for BSPPs can be applied, with the caveat that proximal oracles are required. The standard Euclidean distance has been used in some cases [21], but it requires solving a projection problem that takes $O(n \log^2 n)$ time, where $n$ is the dimension of a player's decision space [16, 21]. While this is "nearly" linear time, such projections have not been used much in practice. Proximal oracles have instead been based on dilated regularizers [23], which lead to a proximal update that can be performed with a single pass over the decision space. With the dilated entropy regularizer, this can be performed in linear time, and this regularizer leads to the strongest bounds on game constants that impact the convergence rate of proximal-oracle-based FOMs [14, 25]. More recently, it has been shown that a specialized *kernelization* can be used to achieve linear-time proximal updates and stronger convergence rates specifically for the dilated entropy with optimistic online mirror descent through a correspondence with optimistic multiplicative weights on the exponentially-many vertices of the decision polytope [6, 17]. Yet this approach was shown to have somewhat disappointing numerical performance in Farina et al. [17], and thus is less important practically despite its theoretical significance.

A completely different approach for first-order-based updates is the CFR framework, which decomposes regret minimization on the EFG decision sets into local simplex-based regret minimization [49]. In theory, CFR-based results have mostly led to an inferior $T^{-1/2}$ rate of convergence, but in practice the CFR framework instantiated with *regret matching*$^+$ (RM$^+$) [41] or *predictive* RM$^+$ (PRM$^+$) [15] is the fastest approach for essentially every EFG setting. RM$^+$ is often fastest for "poker-like" EFGs, while PRM$^+$ is often fastest for other classes of games [15]. Improved rates on the order of $T^{-3/4}$ [12] and $\log T/T$ [4] have been achieved within the CFR framework, but only while using regret minimizers that lead to significantly worse practical performance (in particular, numerically these perform worse than the best $1/T$ FOMs such as mirror prox with appropriate stepsize tuning).

In the last few years there has been a growing literature on *last-iterate convergence* in EFGs. There, the goal is to show that one can converge to an equilibrium without averaging the iterates generated by a FOM or CFR-based method. It has long been known that with the Euclidean regularizer, it is possible to converge at a linear rate in last iterate with e.g., the *extragradient method* (a.k.a. mirror prox with the Euclidean regularizer) on BSPPs with polyhedral decision sets, as they are in EFGs [21, 43, 45]. More recently, it has been shown that a linear rate can be achieved with certain dilated regularizers [27], with the kernelization approach of Farina et al. [17], and in a regularized CFR setup [31]. At this stage, however, these last-iterate results are of greater theoretical significance than practical significance, because the linear rate often does not occur until after quite many iterations, and typically the methods do not match the performance of ergodic methods at reasonable time scales. For this reason, we do not compare to last-iterate algorithms in our experiments.

## B    Treeplex Example

As an example, consider the treeplex of Kuhn poker [26] adapted from [13] shown in Figure 5. Kuhn poker is a game played with a three card deck: jack, queen, and king. In this case, for example, we have $\mathcal{J}_{\mathcal{X}} = \{0, 1, 2, 3, 4, 5, 6\}$, $p_0 = \emptyset$, $p_1 = p_2 = p_3 = (0, \text{start})$, $p_4 = (1, \text{check})$, $p_5 = (2, \text{check})$, $p_6 = (3, \text{check})$, $A_0 = \{\text{start}\}$, $A_1 = A_2 = A_3 = \{check, raise\}$, $A_4 = A_5 = A_6 = \{\text{fold}, \text{call}\}$, $\mathcal{C}_{(0,\text{start})} = \{1, 2, 3\}$, $\mathcal{C}_{(1,\text{raise})} = \mathcal{C}_{(1,\text{raise})} = \mathcal{C}_{(2,\text{raise})} = \mathcal{C}_{(3,\text{raise})} = \mathcal{C}_{(4,\text{fold})} = \mathcal{C}_{(5,\text{fold})} = \mathcal{C}_{(6,\text{fold})} = \mathcal{C}_{(4,\text{call})} = \mathcal{C}_{(5,\text{call})} = \mathcal{C}_{(6,\text{call})} = \emptyset$, $\downarrow 0 = \mathcal{J}_{\mathcal{X}}$, $\downarrow 1 = \{1, 4\}$, $\downarrow 2 = \{2, 5\}$, $\downarrow 3 = \{3, 6\}$, $\downarrow 4 = \{4\}$, $\downarrow 5 = \{5\}$, $\downarrow 6 = \{6\}$. In this case, $\emptyset$ represents the empty sequence.

## C    Proofs

### C.1    Proof of Proposition 3.1

*Proof.* The claim that $\mathbf{x}_k \in \mathcal{X}$, $\mathbf{y}_k \in \mathcal{Y}$ is immediate from the algorithm description, as both are solutions to constrained optimization problems with these same constraints.

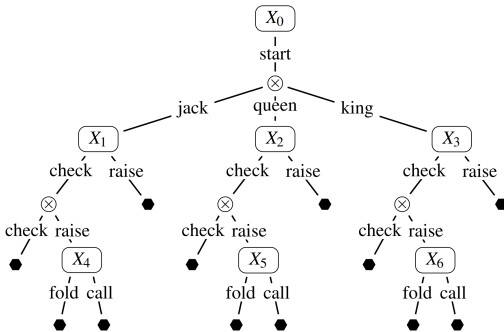

Figure 5: The sequential decision problem for the first player in Kuhn poker. ● represents the end of the decision process and ⊗ represents an observation (which may lead to multiple decision points). Adapted from [13].

For the remaining claim, observe first that

$$\eta_k \langle \mathbf{M}\mathbf{x}_k, \mathbf{v} \rangle = \eta_k \langle \mathbf{g}_k, \mathbf{v} \rangle - D_\psi(\mathbf{v}, \mathbf{y}_{k-1}) + D_\psi(\mathbf{v}, \mathbf{y}_{k-1}) + \eta_k \langle \mathbf{M}\mathbf{x}_k - \mathbf{g}_k, \mathbf{v} \rangle.$$

Recall from Algorithm 1 that

$$\mathbf{y}_k = \operatorname*{argmax}_{\mathbf{v} \in \mathcal{Y}} \left\{ \eta_k \langle \mathbf{g}_k, \mathbf{v} \rangle - D_\psi(\mathbf{v}, \mathbf{y}_{k-1}) \right\}.$$

Define the function under the max defining $\mathbf{y}_k$ by $\Psi_k$. Then as $\Psi_k(\cdot)$ is the sum of $-\psi(\cdot)$ and linear terms, we have $D_{\Psi_k}(\cdot, \mathbf{y}) = -D_\psi(\cdot, \mathbf{y})$, for any $\mathbf{y}$. Further, as $\Psi_k$ is maximized by $\mathbf{y}_k$, we have $\Psi_k(\mathbf{v}) \leq \Psi_k(\mathbf{y}_k) - D_\psi(\mathbf{v}, \mathbf{y}_k)$. Thus, it follows that

$$\begin{aligned} \eta_k \langle \mathbf{M}\mathbf{x}_k, \mathbf{v} \rangle \leq {} & \eta_k \langle \mathbf{g}_k, \mathbf{y}_k \rangle - D_\psi(\mathbf{y}_k, \mathbf{y}_{k-1}) + \langle \mathbf{M}\mathbf{x}_k - \mathbf{g}_k, \mathbf{v} \rangle \\ & - D_\psi(\mathbf{v}, \mathbf{y}_k) + D_\psi(\mathbf{v}, \mathbf{y}_{k-1}). \end{aligned} \tag{C.1}$$

Using the same ideas for the primal side, we have

$$\begin{aligned} \eta_k \langle \mathbf{M}\mathbf{u}, \mathbf{y}_k \rangle \geq {} & \eta_k \langle \mathbf{x}_k, \mathbf{h}_k \rangle + D_\phi(\mathbf{x}_k, \mathbf{x}_{k-1}) + \eta_k \langle \mathbf{u}, \mathbf{M}^\top \mathbf{y}_k - \mathbf{h}_k \rangle \\ & + D_\phi(\mathbf{u}, \mathbf{x}_k) - D_\phi(\mathbf{u}, \mathbf{x}_{k-1}) \end{aligned} \tag{C.2}$$

Combining (C.1) and (C.2),

$$\begin{aligned} \eta_k \mathrm{Gap}^{\mathbf{u},\mathbf{v}}(\mathbf{x}_k, \mathbf{y}_k) \leq {} & \eta_k \langle \mathbf{M}\mathbf{x}_k - \mathbf{g}_k, \mathbf{v} - \mathbf{y}_k \rangle - \eta_k \langle \mathbf{u} - \mathbf{x}_k, \mathbf{M}^\top \mathbf{y}_k - \mathbf{h}_k \rangle \\ & - D_\psi(\mathbf{y}_k, \mathbf{y}_{k-1}) - D_\phi(\mathbf{x}_k, \mathbf{x}_{k-1}) \\ & - D_\phi(\mathbf{u}, \mathbf{x}_k) + D_\phi(\mathbf{u}, \mathbf{x}_{k-1}) - D_\psi(\mathbf{v}, \mathbf{y}_k) + D_\psi(\mathbf{v}, \mathbf{y}_{k-1}). \end{aligned}$$

To complete the proof, it remains to combine the definition of $\mathcal{E}_k$ from the proposition statement with the last inequality. □

## C.2 Proof of Theorem 3.2

For notational convenience, in this proof we define vectors $\hat{\mathbf{x}}_k$ and $\hat{\mathbf{y}}_k$ by $\hat{\mathbf{x}}_k^j = \frac{\mathbf{x}_k^j}{x_k^{p_j}} x_{k-1}^{p_j}$ for $j \in \mathcal{J}_{\mathcal{X}}$ and $\hat{\mathbf{y}}_k^j = \frac{\mathbf{y}_k^j}{y_k^{p_j}} y_{k-1}^{p_j}$ for $j \in \mathcal{J}_{\mathcal{Y}}$, so that $\widetilde{\mathbf{x}}_k = \mathbf{x}_k - \hat{\mathbf{x}}_k - \frac{\eta_{k-1}}{\eta_k}(\mathbf{x}_{k-1} - \hat{\mathbf{x}}_{k-1})$ and $\widetilde{\mathbf{y}}_k = \mathbf{y}_k - \hat{\mathbf{y}}_k - \frac{\eta_{k-1}}{\eta_k}(\mathbf{y}_{k-1} - \hat{\mathbf{y}}_{k-1})$.

To prove the theorem, we first prove the following auxiliary lemma which bounds the inner product terms appearing in the error terms $\mathcal{E}_k$.

**Lemma C.1.** *In all iterations $k$ of Algorithm 1, for any $(\mathbf{u}, \mathbf{v}) \in \mathcal{X} \times \mathcal{Y}$ and any $\alpha, \beta > 0$,*

$$\begin{aligned} \eta_k \langle \mathbf{M}\mathbf{x}_k - \mathbf{g}_k, \mathbf{v} - \mathbf{y}_k \rangle \leq {} & \eta_k \langle \mathbf{M}_{\mathbf{y}}(\mathbf{x}_k - \hat{\mathbf{x}}_k), \mathbf{v} - \mathbf{y}_k \rangle - \eta_{k-1} \langle \mathbf{M}_{\mathbf{y}}(\mathbf{x}_{k-1} - \hat{\mathbf{x}}_{k-1}), \mathbf{v} - \mathbf{y}_{k-1} \rangle \\ & + \eta_k \langle \mathbf{M}_{\mathbf{x}}(\mathbf{x}_k - \mathbf{x}_{k-1}), \mathbf{v} - \mathbf{y}_k \rangle - \eta_{k-1} \langle \mathbf{M}_{\mathbf{x}}(\mathbf{x}_{k-1} - \mathbf{x}_{k-2}), \mathbf{v} - \mathbf{y}_{k-1} \rangle \\ & + \eta_{k-1} \frac{\|\mathbf{M}_{\mathbf{x}}\|_* + \|\mathbf{M}_{\mathbf{y}}\|_*}{2} \left( \alpha \|\mathbf{x}_{k-1} - \mathbf{x}_{k-2}\|^2 + \frac{1}{\alpha} \|\mathbf{y}_k - \mathbf{y}_{k-1}\|^2 \right). \end{aligned}$$

*and*

$$-\eta_k \left\langle \mathbf{u} - \mathbf{x}_k, \mathbf{M}^\top \mathbf{y}_k - \mathbf{h}_k \right\rangle \leq - \eta_k \left\langle \mathbf{M}_\mathbf{x}^\top (\mathbf{y}_k - \hat{\mathbf{y}}_k), \mathbf{u} - \mathbf{x}_k \right\rangle + \eta_{k-1} \left\langle \mathbf{M}_\mathbf{x}^\top (\mathbf{y}_{k-1} - \hat{\mathbf{y}}_{k-1}), \mathbf{u} - \mathbf{x}_{k-1} \right\rangle$$
$$- \eta_k \left\langle \mathbf{M}_\mathbf{y}^\top (\mathbf{y}_k - \mathbf{y}_{k-1}), \mathbf{u} - \mathbf{x}_k \right\rangle + \eta_{k-1} \left\langle \mathbf{M}_\mathbf{y}^\top (\mathbf{y}_{k-1} - \mathbf{y}_{k-2}), \mathbf{u} - \mathbf{x}_{k-1} \right\rangle$$
$$+ \eta_{k-1} \frac{\|\mathbf{M}_\mathbf{x}^\top + \mathbf{M}_\mathbf{y}^\top\|_*}{2} \left( \beta \|\mathbf{x}_{k-1} - \mathbf{x}_k\|^2 + \frac{1}{\beta} \|\mathbf{y}_{k-1} - \mathbf{y}_{k-2}\|^2 \right).$$

*Proof.* Observe first that, by Algorithm 1,

$$\mathbf{M}^{(t,:)} \mathbf{x}_k - \mathbf{g}_k^{(t)} = \mathbf{M}^{(t,1:t)} \left( \mathbf{x}_k - \hat{\mathbf{x}}_k - \frac{\eta_{k-1}}{\eta_k} (\mathbf{x}_{k-1} - \hat{\mathbf{x}}_{k-1}) \right)^{(1:t)}$$
$$\qquad\qquad + \mathbf{M}^{(t,t+1:s)} \left( \mathbf{x}_k - \mathbf{x}_{k-1} - \frac{\eta_{k-1}}{\eta_k} (\mathbf{x}_{k-1} - \mathbf{x}_{k-2}) \right)^{(t+1:s)}. \tag{C.3}$$

Additionally, by definition (see Eq. (3.1)), $\sum_{t=1}^s \mathbf{M}_{t,1:t} = \mathbf{M} - \mathbf{M}_\mathbf{x} = \mathbf{M}_\mathbf{y}$. Hence,

$$\eta_k \sum_{t=1}^s \left\langle \mathbf{M}^{(t,1:t)} \left( \mathbf{x}_k - \hat{\mathbf{x}}_k - \frac{\eta_{k-1}}{\eta_k} (\mathbf{x}_{k-1} - \hat{\mathbf{x}}_{k-1}) \right)^{(1:t)}, \mathbf{v}^{(t)} - \mathbf{y}_k^{(t)} \right\rangle$$
$$= \eta_k \sum_{t=1}^s \left\langle \mathbf{M}_{t,1:t} \left( \mathbf{x}_k - \hat{\mathbf{x}}_k - \frac{\eta_{k-1}}{\eta_k} (\mathbf{x}_{k-1} - \hat{\mathbf{x}}_{k-1}) \right), \mathbf{v} - \mathbf{y}_k \right\rangle$$
$$= \eta_k \left\langle \mathbf{M}_\mathbf{y} \left( \mathbf{x}_k - \hat{\mathbf{x}}_k - \frac{\eta_{k-1}}{\eta_k} (\mathbf{x}_{k-1} - \hat{\mathbf{x}}_{k-1}) \right), \mathbf{v} - \mathbf{y}_k \right\rangle$$
$$= \eta_k \left\langle \mathbf{M}_\mathbf{y} (\mathbf{x}_k - \hat{\mathbf{x}}_k), \mathbf{v} - \mathbf{y}_k \right\rangle - \eta_{k-1} \left\langle \mathbf{M}_\mathbf{y} (\mathbf{x}_{k-1} - \hat{\mathbf{x}}_{k-1}), \mathbf{v} - \mathbf{y}_{k-1} \right\rangle$$
$$\qquad - \eta_{k-1} \left\langle \mathbf{M}_\mathbf{y} (\mathbf{x}_{k-1} - \hat{\mathbf{x}}_{k-1}), \mathbf{y}_{k-1} - \mathbf{y}_k \right\rangle. \tag{C.4}$$

The first two terms in (C.4) telescope, so we focus on bounding $-\eta_{k-1} \left\langle \mathbf{M}_\mathbf{y} (\mathbf{x}_{k-1} - \hat{\mathbf{x}}_{k-1}), \mathbf{y}_{k-1} - \mathbf{y}_k \right\rangle$. By definition of $\hat{\mathbf{x}}_k$, for all $j \in \mathcal{J}_\mathcal{X}$,

$$\mathbf{x}_{k-1}^j - \hat{\mathbf{x}}_{k-1}^j = \frac{\mathbf{x}_{k-1}^j}{x_{k-1}^{p_j}} \left( x_{k-1}^{p_j} - x_{k-2}^{p_j} \right).$$

By the definition of a treeplex, each vector $\frac{\mathbf{x}_{k-1}^j}{x_{k-1}^{p_j}}$ belongs to a probability simplex of the appropriate size. This further implies that

$$\|\mathbf{x}_{k-1} - \hat{\mathbf{x}}_{k-1}\| = \left\| \left[ \frac{\mathbf{x}_{k-1}^j}{x_{k-1}^{p_j}} \left( x_{k-1}^{p_j} - x_{k-2}^{p_j} \right) \right]_{j \in \mathcal{J}_\mathcal{X}} \right\|$$
$$\leq \| [x_{k-1}^{p_j} - x_{k-2}^{p_j}]_{j \in \mathcal{J}_\mathcal{X}} \|$$
$$\leq \|\mathbf{x}_{k-1} - \mathbf{x}_{k-2}\|, \tag{C.5}$$

where the notation $[a^j]_{j \in \mathcal{J}_\mathcal{X}}$ is used to denote the vector with entries $a^j$, for $j \in \mathcal{J}_\mathcal{X}$. The first inequality in (C.5) holds for any $\ell_p$ norm ($p \geq 1$), by its definition and $\left\langle \mathbf{1}, \mathbf{x}_{k-1}^j \right\rangle = 1$, $\forall j$. Thus, applying the definitions of the norms from the preliminaries,

$$- \left\langle \mathbf{M}_\mathbf{y} (\mathbf{x}_{k-1} - \hat{\mathbf{x}}_{k-1}), \mathbf{y}_{k-1} - \mathbf{y}_k \right\rangle \leq \|\mathbf{M}_\mathbf{y} (\mathbf{x}_{k-1} - \hat{\mathbf{x}}_{k-1})\|_* \|\mathbf{y}_{k-1} - \mathbf{y}_k\|$$
$$\leq \|\mathbf{M}_\mathbf{y}\|_* \|\mathbf{x}_{k-1} - \mathbf{x}_{k-2}\| \|\mathbf{y}_{k-1} - \mathbf{y}_k\|$$
$$\leq \frac{\|\mathbf{M}_\mathbf{y}\|_*}{2} \left( \alpha \|\mathbf{x}_{k-1} - \mathbf{x}_{k-2}\|^2 + \frac{1}{\alpha} \|\mathbf{y}_k - \mathbf{y}_{k-1}\|^2 \right), \tag{C.6}$$

where the last line is by Young's inequality and holds for any $\alpha > 0$.

On the other hand, recalling that $\mathbf{M_x} = \sum_{t=1}^{s-1} \mathbf{M}_{t,t+1:s}$, we also have

$$\eta_k \sum_{t=1}^{s} \left\langle \mathbf{M}^{(t,t+1:s)} \Big( \mathbf{x}_k - \mathbf{x}_{k-1} - \frac{\eta_{k-1}}{\eta_k}(\mathbf{x}_{k-1} - \mathbf{x}_{k-2}) \Big)^{(t+1:s)}, \mathbf{v}^{(t)} - \mathbf{y}_k^{(t)} \right\rangle$$

$$= \eta_k \sum_{t=1}^{s} \left\langle \mathbf{M}_{t,t+1:s} \Big( \mathbf{x}_k - \mathbf{x}_{k-1} - \frac{\eta_{k-1}}{\eta_k}(\mathbf{x}_{k-1} - \mathbf{x}_{k-2}) \Big), \mathbf{v} - \mathbf{y}_k \right\rangle$$

$$= \eta_k \left\langle \mathbf{M_x} \Big( \mathbf{x}_k - \mathbf{x}_{k-1} - \frac{\eta_{k-1}}{\eta_k}(\mathbf{x}_{k-1} - \mathbf{x}_{k-2}) \Big), \mathbf{v} - \mathbf{y}_k \right\rangle$$

$$= \eta_k \langle \mathbf{M_x}(\mathbf{x}_k - \mathbf{x}_{k-1}), \mathbf{v} - \mathbf{y}_k \rangle - \eta_{k-1} \langle \mathbf{M_x}(\mathbf{x}_{k-1} - \mathbf{x}_{k-2}), \mathbf{v} - \mathbf{y}_k \rangle$$

$$= \eta_k \langle \mathbf{M_x}(\mathbf{x}_k - \mathbf{x}_{k-1}), \mathbf{v} - \mathbf{y}_k \rangle - \eta_{k-1} \langle \mathbf{M_x}(\mathbf{x}_{k-1} - \mathbf{x}_{k-2}), \mathbf{v} - \mathbf{y}_{k-1} \rangle$$

$$\qquad + \eta_{k-1} \langle \mathbf{M_x}(\mathbf{x}_{k-1} - \mathbf{x}_{k-2}), \mathbf{y}_k - \mathbf{y}_{k-1} \rangle. \tag{C.7}$$

Observe that in (C.7) the first two terms telescope and thus we only need to focus on bounding the last term. Applying the definitions of dual and matrix norms from Section 2 and using Young's inequality, we have that for any $\alpha > 0$,

$$\langle \mathbf{M_x}(\mathbf{x}_{k-1} - \mathbf{x}_{k-2}), \mathbf{y}_k - \mathbf{y}_{k-1} \rangle \leq \|\mathbf{M_x}(\mathbf{x}_{k-1} - \mathbf{x}_{k-2})\|_* \|\mathbf{y}_k - \mathbf{y}_{k-1}\|$$

$$\leq \|\mathbf{M_x}\|_* \|\mathbf{x}_{k-1} - \mathbf{x}_{k-2}\| \|\mathbf{y}_k - \mathbf{y}_{k-1}\|$$

$$\leq \frac{\|\mathbf{M_x}\|_*}{2} \Big( \alpha \|\mathbf{x}_{k-1} - \mathbf{x}_{k-2}\|^2 + \frac{1}{\alpha} \|\mathbf{y}_k - \mathbf{y}_{k-1}\|^2 \Big). \tag{C.8}$$

Hence, combining (C.3)–(C.8), we can conclude that

$$\eta_k \langle \mathbf{Mx}_k - \mathbf{g}_k, \mathbf{v} - \mathbf{y}_k \rangle \leq \eta_k \langle \mathbf{M_y}(\mathbf{x}_k - \hat{\mathbf{x}}_k), \mathbf{v} - \mathbf{y}_k \rangle - \eta_{k-1} \langle \mathbf{M_y}(\mathbf{x}_{k-1} - \hat{\mathbf{x}}_{k-1}), \mathbf{v} - \mathbf{y}_{k-1} \rangle$$

$$+ \eta_k \langle \mathbf{M_x}(\mathbf{x}_k - \mathbf{x}_{k-1}), \mathbf{v} - \mathbf{y}_k \rangle - \eta_{k-1} \langle \mathbf{M_x}(\mathbf{x}_{k-1} - \mathbf{x}_{k-2}), \mathbf{v} - \mathbf{y}_{k-1} \rangle$$

$$+ \eta_{k-1} \frac{\|\mathbf{M_x}\|_* + \|\mathbf{M_y}\|_*}{2} \Big( \alpha \|\mathbf{x}_{k-1} - \mathbf{x}_{k-2}\|^2 + \frac{1}{\alpha} \|\mathbf{y}_k - \mathbf{y}_{k-1}\|^2 \Big),$$

completing the proof of the first claim.

Similarly, we observe from Algorithm 1 that

$$\big( \mathbf{M}^{(:,t)} \big)^\top \mathbf{y}_k - \mathbf{h}_k^{(t)} = \big( \mathbf{M}^{(1:t-1,t)} \big)^\top \Big( \mathbf{y}_k - \hat{\mathbf{y}}_k + \frac{\eta_{k-1}}{\eta_k}(\mathbf{y}_{k-1} - \hat{\mathbf{y}}_{k-1}) \Big)^{(1:t-1)}$$

$$+ \big( \mathbf{M}^{(t:s,t)} \big)^\top \Big( \mathbf{y}_k - \mathbf{y}_{k-1} + \frac{\eta_{k-1}}{\eta_k}(\mathbf{y}_{k-1} - \mathbf{y}_{k-2}) \Big)^{(t:s)}.$$

Observing that $\sum_{t=1}^{s} \mathbf{M}_{1:t-1,t} = \mathbf{M_x}$ and $\sum_{t=1}^{s} \mathbf{M}_{t:s,t} = \mathbf{M_y}$, using the same sequence of arguments as for bounding (C.3), we can conclude that for any $\beta > 0$,

$$- \eta_k \langle \mathbf{u} - \mathbf{x}_k, \mathbf{M}^\top \mathbf{y}_k - \mathbf{h}_k \rangle$$

$$\leq - \eta_k \langle \mathbf{M_x}^\top(\mathbf{y}_k - \hat{\mathbf{y}}_k), \mathbf{u} - \mathbf{x}_k \rangle + \eta_{k-1} \langle \mathbf{M_x}^\top(\mathbf{y}_{k-1} - \hat{\mathbf{y}}_{k-1}), \mathbf{u} - \mathbf{x}_{k-1} \rangle$$

$$- \eta_k \langle \mathbf{M_y}^\top(\mathbf{y}_k - \mathbf{y}_{k-1}), \mathbf{u} - \mathbf{x}_k \rangle + \eta_{k-1} \langle \mathbf{M_y}^\top(\mathbf{y}_{k-1} - \mathbf{y}_{k-2}), \mathbf{u} - \mathbf{x}_{k-1} \rangle$$

$$+ \eta_{k-1} \frac{\|\mathbf{M_x}^\top + \mathbf{M_y}^\top\|_*}{2} \Big( \beta \|\mathbf{x}_{k-1} - \mathbf{x}_k\|^2 + \frac{1}{\beta} \|\mathbf{y}_{k-1} - \mathbf{y}_{k-2}\|^2 \Big),$$

completing the proof. $\qquad \square$

*Proof Theorem 3.2.* Recalling the definition of $\mathcal{E}_k$, by Lemma C.1,

$$\mathcal{E}_k \leq \eta_k \langle \mathbf{M_y}(\mathbf{x}_k - \hat{\mathbf{x}}_k), \mathbf{v} - \mathbf{y}_k \rangle - \eta_{k-1} \langle \mathbf{M_y}(\mathbf{x}_{k-1} - \hat{\mathbf{x}}_{k-}), \mathbf{v} - \mathbf{y}_{k-1} \rangle$$

$$+ \eta_k \langle \mathbf{M_x}(\mathbf{x}_k - \mathbf{x}_{k-1}), \mathbf{v} - \mathbf{y}_k \rangle - \eta_{k-1} \langle \mathbf{M_x}(\mathbf{x}_{k-1} - \mathbf{x}_{k-2}), \mathbf{v} - \mathbf{y}_{k-1} \rangle$$

$$+ \eta_{k-1} \frac{\|\mathbf{M_x}\|_* + \|\mathbf{M_y}\|_*}{2} \Big( \alpha \|\mathbf{x}_{k-1} - \mathbf{x}_{k-2}\|^2 + \frac{1}{\alpha} \|\mathbf{y}_k - \mathbf{y}_{k-1}\|^2 \Big)$$

$$- \eta_k \langle \mathbf{M_x}^\top(\mathbf{y}_k - \hat{\mathbf{y}}_k), \mathbf{u} - \mathbf{x}_k \rangle + \eta_{k-1} \langle \mathbf{M_x}^\top(\mathbf{y}_{k-1} - \hat{\mathbf{y}}_{k-1}), \mathbf{u} - \mathbf{x}_{k-1} \rangle \tag{C.9}$$

$$- \eta_k \langle \mathbf{M_y}^\top(\mathbf{y}_k - \mathbf{y}_{k-1}), \mathbf{u} - \mathbf{x}_k \rangle + \eta_{k-1} \langle \mathbf{M_y}^\top(\mathbf{y}_{k-1} - \mathbf{y}_{k-2}), \mathbf{u} - \mathbf{x}_{k-1} \rangle$$

$$+ \eta_{k-1} \frac{\|\mathbf{M_x}^\top\|_* + \|\mathbf{M_y}^\top\|_*}{2} \Big( \beta \|\mathbf{x}_{k-1} - \mathbf{x}_k\|^2 + \frac{1}{\beta} \|\mathbf{y}_{k-1} - \mathbf{y}_{k-2}\|^2 \Big)$$

$$- D_\psi(\mathbf{y}_k, \mathbf{y}_{k-1}) - D_\phi(\mathbf{x}_k, \mathbf{x}_{k-1}).$$

Recalling that $\psi$ is $c_\psi$-strongly convex, $\phi$ is $c_\phi$-strongly convex, setting $\alpha = \beta = \sqrt{\frac{c_\phi}{c_\psi}}$, and using that $\eta_{k-1} \le \frac{\sqrt{c_\phi c_\psi}}{\|\mathbf{M}_\mathbf{x}\|_* + \|\mathbf{M}_\mathbf{y}\|_* + \|\mathbf{M}_\mathbf{x}^\top\|_* + \|\mathbf{M}_\mathbf{y}^\top\|_*} = \frac{\sqrt{c_\phi c_\psi}}{\mu_x + \mu_y}$, (C.9) simplifies to

$$
\begin{aligned}
\mathcal{E}_k \le\; & \eta_k \left\langle \mathbf{M}_\mathbf{y}(\mathbf{x}_k - \hat{\mathbf{x}}_k), \mathbf{v} - \mathbf{y}_k \right\rangle - \eta_{k-1} \left\langle \mathbf{M}_\mathbf{y}(\mathbf{x}_{k-1} - \hat{\mathbf{x}}_{k-}), \mathbf{v} - \mathbf{y}_{k-1} \right\rangle \\
& + \eta_k \left\langle \mathbf{M}_\mathbf{x}(\mathbf{x}_k - \mathbf{x}_{k-1}), \mathbf{v} - \mathbf{y}_k \right\rangle - \eta_{k-1} \left\langle \mathbf{M}_\mathbf{x}(\mathbf{x}_{k-1} - \mathbf{x}_{k-2}), \mathbf{v} - \mathbf{y}_{k-1} \right\rangle \\
& - \eta_k \left\langle \mathbf{M}_\mathbf{x}^\top(\mathbf{y}_k - \hat{\mathbf{y}}_k), \mathbf{u} - \mathbf{x}_k \right\rangle + \eta_{k-1} \left\langle \mathbf{M}_\mathbf{x}^\top(\mathbf{y}_{k-1} - \hat{\mathbf{y}}_{k-1}), \mathbf{u} - \mathbf{x}_{k-1} \right\rangle \\
& - \eta_k \left\langle \mathbf{M}_\mathbf{y}^\top(\mathbf{y}_k - \mathbf{y}_{k-1}), \mathbf{u} - \mathbf{x}_k \right\rangle + \eta_{k-1} \left\langle \mathbf{M}_\mathbf{y}^\top(\mathbf{y}_{k-1} - \mathbf{y}_{k-2}), \mathbf{u} - \mathbf{x}_{k-1} \right\rangle \\
& + \frac{c_\psi \mu_y}{2(\mu_x + \mu_y)} \left( \|\mathbf{y}_{k-1} - \mathbf{y}_{k-2}\|^2 - \|\mathbf{y}_k - \mathbf{y}_{k-1}\|^2 \right) \\
& + \frac{c_\phi \mu_x}{2(\mu_x + \mu_y)} \left( \|\mathbf{x}_{k-1} - \mathbf{x}_{k-2}\|^2 - \|\mathbf{x}_k - \mathbf{x}_{k-1}\|^2 \right).
\end{aligned}
\tag{C.10}
$$

Telescoping (C.10) and recalling that $\eta_0 = 0$, we now have

$$
\begin{aligned}
\sum_{k=1}^K \mathcal{E}_k \le\; & \eta_K \left\langle \mathbf{M}_\mathbf{x}(\mathbf{x}_K - \mathbf{x}_{K-1}), \mathbf{v} - \mathbf{y}_K \right\rangle - \eta_K \left\langle \mathbf{M}_\mathbf{y}^\top(\mathbf{y}_K - \mathbf{y}_{K-1}), \mathbf{u} - \mathbf{x}_K \right\rangle \\
& + \eta_K \left\langle \mathbf{M}_\mathbf{y}(\mathbf{x}_K - \hat{\mathbf{x}}_K), \mathbf{v} - \mathbf{y}_K \right\rangle - \eta_K \left\langle \mathbf{M}_\mathbf{x}^\top(\mathbf{y}_K - \hat{\mathbf{y}}_K), \mathbf{u} - \mathbf{x}_K \right\rangle \\
& - \frac{c_\psi \mu_y}{2(\mu_x + \mu_y)} \|\mathbf{y}_K - \mathbf{y}_{K-1}\|^2 \\
& - \frac{c_\phi \mu_x}{2(\mu_x + \mu_y)} \|\mathbf{x}_K - \mathbf{x}_{K-1}\|^2.
\end{aligned}
\tag{C.11}
$$

Observe that $\mathrm{Gap}^{\mathbf{u},\mathbf{v}}(\cdot, \cdot)$ is linear in both its arguments. Hence, $\mathrm{Gap}^{\mathbf{u},\mathbf{v}}(\bar{\mathbf{x}}_K, \bar{\mathbf{y}}_K) = \frac{1}{H_K} \sum_{k=1}^K \eta_k \mathrm{Gap}^{\mathbf{u},\mathbf{v}}(\mathbf{x}_k, \mathbf{y}_k)$. Applying Proposition 3.1 and combining with (C.11), we now have

$$
\begin{aligned}
H_K \mathrm{Gap}^{\mathbf{u},\mathbf{v}}(\bar{\mathbf{x}}_K, \bar{\mathbf{y}}_K) \le\; & D_\phi(\mathbf{u}, \mathbf{x}_0) + D_\psi(\mathbf{v}, \mathbf{y}_0) \\
& + \eta_K \left\langle \mathbf{M}_\mathbf{x}(\mathbf{x}_K - \mathbf{x}_{K-1}), \mathbf{v} - \mathbf{y}_K \right\rangle - \eta_K \left\langle \mathbf{M}_\mathbf{y}^\top(\mathbf{y}_K - \mathbf{y}_{K-1}), \mathbf{u} - \mathbf{x}_K \right\rangle \\
& + \eta_K \left\langle \mathbf{M}_\mathbf{y}(\mathbf{x}_K - \hat{\mathbf{x}}_K), \mathbf{v} - \mathbf{y}_K \right\rangle - \eta_K \left\langle \mathbf{M}_\mathbf{x}^\top(\mathbf{y}_K - \hat{\mathbf{y}}_K), \mathbf{u} - \mathbf{x}_K \right\rangle \\
& - \frac{c_\psi \mu_y}{2(\mu_x + \mu_y)} \|\mathbf{y}_K - \mathbf{y}_{K-1}\|^2 - \frac{c_\phi \mu_x}{2(\mu_x + \mu_y)} \|\mathbf{x}_K - \mathbf{x}_{K-1}\|^2 \\
& - D_\phi(\mathbf{u}, \mathbf{x}_K) - D_\psi(\mathbf{v}, \mathbf{y}_K).
\end{aligned}
\tag{C.12}
$$

To complete bounding the gap, it remains to argue that the right-hand side of (C.12) is bounded by $D_\phi(\mathbf{u}, \mathbf{x}_0) + D_\psi(\mathbf{v}, \mathbf{y}_0) - \frac{\mu_x}{\mu_x + \mu_y} D_\phi(\mathbf{u}, \mathbf{x}_K) - \frac{\mu_y}{\mu_x + \mu_y} D_\psi(\mathbf{v}, \mathbf{y}_K)$. This is done using the same sequence of arguments as in bounding $\mathcal{E}_k$ and is omitted.

Let $(\mathbf{x}^*, \mathbf{y}^*) \in \mathcal{X} \times \mathcal{Y}$ be any primal-dual solution to (PD). Then $\mathrm{Gap}^{(\mathbf{x}^*, \mathbf{y}^*)}(\bar{\mathbf{x}}_K, \bar{\mathbf{y}}_K) \ge 0$ and we can conclude that

$$
\frac{\mu_x}{\mu_x + \mu_y} D_\phi(\mathbf{x}^*, \mathbf{x}_K) + \frac{\mu_y}{\mu_x + \mu_y} D_\psi(\mathbf{y}^*, \mathbf{y}_K) \le D_\phi(\mathbf{x}^*, \mathbf{x}_0) + D_\psi(\mathbf{y}^*, \mathbf{y}_0).
$$

Further, using that $D_\phi(\cdot, \cdot) \ge 0$, $D_\psi(\cdot, \cdot) \ge 0$, we can also conclude that

$$
\begin{aligned}
\sup_{\mathbf{u} \in \mathcal{X}, \mathbf{v} \in \mathcal{Y}} \mathrm{Gap}^{\mathbf{u},\mathbf{v}}(\bar{\mathbf{x}}_K, \bar{\mathbf{y}}_K) &= \sup_{\mathbf{u} \in \mathcal{X}, \mathbf{v} \in \mathcal{Y}} \{ \langle \mathbf{M} \bar{\mathbf{x}}_K, \mathbf{v} \rangle - \langle \mathbf{M} \mathbf{u}, \bar{\mathbf{y}}_K \rangle \} \\
&\le \frac{\sup_{\mathbf{u} \in \mathcal{X}, \mathbf{v} \in \mathcal{Y}} \{ D_\phi(\mathbf{u}, \mathbf{x}_0) + D_\psi(\mathbf{v}, \mathbf{y}_0) \}}{H_K}.
\end{aligned}
$$

Finally, setting $\eta_k = \frac{\sqrt{c_\phi c_\psi}}{\mu_x + \mu_y}$ for all $k \ge 1$ immediately leads to the conclusion that $H_K = K \frac{\sqrt{c_\phi c_\psi}}{\mu_x + \mu_y}$, as $H_K = \sum_{k=1}^K \eta_k$, by definition. The last bound is by setting $\frac{\sup_{\mathbf{u} \in \mathcal{X}, \mathbf{v} \in \mathcal{Y}} \{ D_\phi(\mathbf{u}, \mathbf{x}_0) + D_\psi(\mathbf{v}, \mathbf{y}_0) \}}{H_K} \le \epsilon$, and solving for $K$. $\qquad\square$

## D Algorithm Implementation Details

In Algorithm 2, we present an implementation-specific version of ECyclicPDA, in order to make it clear that our algorithm can be implemented without any extra computation compared to the

computation needed for gradient and prox updates in MP. Note that MP performs two gradient computations and two prox computations per player, due to how it achieves "extrapolation"; we want to argue that we perform an equivalent number operations as needed for a *single* gradient computation and prox computation per player. Note that the overall complexity of first-order methods when applied to EFGs is dominated by the gradient and prox update computations; this is why we compare our algorithm to MP on this basis. The key differences from Algorithm 1 are that we explicitly use $\hat{\mathbf{x}}_k$ and $\hat{\mathbf{y}}_k$ to represent the behavioral strategy that is computed via the partial prox updates (which are then scaled at the end of a full iteration of our method to $\mathbf{x}_k$ and $\mathbf{y}_k$), and that we use $\hat{\mathbf{h}}_k^j$ and $\hat{\mathbf{g}}_k^j$ to accumulate gradient contributions from decision points that occur underneath $j$, to make the partial prox update explicit.

In Lines 8 and 13, we are only dealing with the columns and rows, respectively, of the payoff matrix that correspond to the current block number $t$, which means that as $t$ ranges from 1 to $s$, for the computation of the gradient, we will only consider each column and row, respectively, once, as would have to be done in a full gradient computation for MP.

The more difficult aspect of the implementation is ensuring that we do the same number of operations for the prox computation in ECyclicPDA as an analogous single prox computation in MP. We achieve this by applying the updates in Proposition 2.2 only for the decision points in the current block, in Lines 9 to 12 for $\mathbf{x}$ and 14 to 17 for $\mathbf{y}$.

We focus on the updates for $\mathbf{x}$; the argument is analogous for $\mathbf{y}$. When applying this local prox update for decision point $j \in \mathcal{J}_{\mathcal{X}}^{(t)}$, we have already correctly computed $\hat{\mathbf{h}}^j$, the contributions to the gradient for the local prox update that originate from the children of $j$, again because the blocks represent the treeplex ordering; in particular, whenever we have encountered a child decision point of $j$ in the past, we accumulate its contribution to the gradient for its parent at $\hat{\mathbf{h}}^{p_j}$. Since the prox update decomposition from Proposition 2.2 has to be applied for every single decision point in $\mathcal{J}_{\mathcal{X}}$ in a full prox update (as done in MP), we again do not incur any dependence on the number of blocks.

## E   Description of EFG Benchmarks

We provide game descriptions of the games we run our experiments on below. Our game descriptions are adapted from Farina et al. [15]. In Table 2, we provide the number of sequences for player $\mathbf{x}$ ($n$), the number of sequences for player $\mathbf{y}$ ($m$), and the number of leaves in the game (NNZ($\mathbf{M}$)).

Table 2: Number of sequences for both players and number of leaves for each game. These correspond to the dimensions $n$ and $m$ of $\mathbf{M}$, and the number of nonzero entries of $\mathbf{M}$, respectively.

| Game | Num. of $\mathbf{x}$ sequences | Num. of $\mathbf{y}$ sequences | Num. of leaves |
|---|---|---|---|
| Goofspiel (4 ranks) | 21329 | 21329 | 13824 |
| Liar's Dice | 24571 | 24571 | 147420 |
| Leduc (13 ranks) | 6007 | 6007 | 98956 |
| Battleship | 73130 | 253940 | 552132 |

### E.1   Goofspiel (4 ranks)

Goofspiel is a card-based game that is a standard benchmark in the EFG-solving community [36]. In the version that we test on, there are 4 unique cards (ranks), and there are 3 copies of each rank, divided into 3 separate decks. Each player gets a deck, and the third deck is known as the prize deck. Cards are randomly drawn from the prize deck, and each player submits a bid for the drawn card by submitting a card from one of their respective decks, the value of which represents their bid. Whoever submits the higher bid wins the card from the prize deck. Once all the cards from the prize deck have been drawn, bid on, and won by one of the players, the game terminates, and the payoffs for players are given by the sum of the prize cards they won.

---

**Algorithm 2** Extrapolated Cyclic Primal-Dual EFG Solver (Implementation Version)

---

1: **Input:** $\mathbf{M}, m, n$

2: **Initialization:** $\mathbf{x}_0 \in \mathcal{X}, \mathbf{y}_0 \in \mathcal{Y}, \eta_0 = H_0 = 0, \eta = \frac{\sqrt{c_\phi c_\psi}}{\mu_x + \mu_y}, \bar{\mathbf{x}}_0 = \mathbf{x}_0, \bar{\mathbf{y}}_0 = \mathbf{y}_0, \mathbf{g}_0 = \mathbf{0}, \mathbf{h}_0 = \mathbf{0}$

3: **for** $k = 1 : K$ **do**

4:      Choose $\eta_k \leq \eta$, $H_k = H_{k-1} + \eta_k$

5:      $\mathbf{g}_k = \mathbf{0}, \mathbf{h}_k = \mathbf{0}, \hat{\mathbf{g}}_k = \mathbf{0}, \hat{\mathbf{h}}_k = \mathbf{0}$

6:      $\widetilde{\mathbf{x}}_k = \mathbf{x}_{k-1} + \frac{\eta_{k-1}}{\eta_k}(\mathbf{x}_{k-1} - \mathbf{x}_{k-2}), \widetilde{\mathbf{y}}_k = \mathbf{y}_{k-1} + \frac{\eta_{k-1}}{\eta_k}(\mathbf{y}_{k-1} - \mathbf{y}_{k-2})$

7:      **for** $t = 1 : s$ **do**

8:          $\mathbf{h}_k^{(t)} = (\mathbf{M}^{(:,t)})^\top \widetilde{\mathbf{y}}_k$

9:          **for** $j \in \mathcal{J}_{\mathcal{X}}^{(t)}$ **do**

10:              $\hat{\mathbf{x}}_k^j = \mathrm{argmin}_{\mathbf{b}^j \in \Delta^{n_j}} \left\{ \left\langle \hat{\mathbf{h}}_k^j + \mathbf{h}_k^j, \mathbf{b}^j \right\rangle + D_{\phi^j}\left(\mathbf{b}^j, \hat{\mathbf{x}}_{k-1}^j\right) \right\}$

11:              $(j', a) = p_j$

12:              $\hat{h}_k^{p_j} \mathrel{+}= \left[ \phi^{\downarrow j^*}\left(-\mathbf{h}_k^{\downarrow j} + \nabla\phi^{\downarrow j}\left(x_{k-1}^{\downarrow j}\right)\right) - \phi^j\left(\hat{\mathbf{x}}_{k-1}^j\right) + \left\langle \nabla\phi^j\left(\hat{\mathbf{x}}_{k-1}^j\right), \hat{\mathbf{x}}_{k-1}^j \right\rangle \right]$

13:          $\mathbf{g}_k^{(t)} = \mathbf{M}^{(t,:)} \widetilde{\mathbf{x}}_k$

14:          **for** $j \in \mathcal{J}_{\mathcal{Y}}^{(t)}$ **do**

15:              $\hat{\mathbf{y}}_k^j = \mathrm{argmin}_{\mathbf{b}^j \in \Delta^{n_j}} \left\{ \left\langle \hat{\mathbf{g}}_k^j + \mathbf{g}_k^j, \mathbf{b}^j \right\rangle + D_{\psi^j}\left(\mathbf{b}^j, \hat{\mathbf{y}}_{k-1}^j\right) \right\}$

16:              $(j', a) = p_j$

17:              $\hat{h}_k^{p_j} \mathrel{+}= \left[ \psi^{\downarrow j^*}\left(-\mathbf{g}_k^{\downarrow j} + \nabla\psi^{\downarrow j}\left(y_{k-1}^{\downarrow j}\right)\right) - \psi^j\left(\hat{\mathbf{y}}_{k-1}^j\right) + \left\langle \nabla\psi^j\left(\hat{\mathbf{y}}_{k-1}^j\right), \hat{\mathbf{y}}_{k-1}^j \right\rangle \right]$

18:      **for** $j \in \mathcal{J}_{\mathcal{X}}^{(t)}$ **do**

19:          $\widetilde{\mathbf{x}}_k^j = \left[ \hat{\mathbf{x}}_k^j x_{k-1}^{p_j} + \frac{\eta_{k-1}}{\eta_k}\left(\mathbf{x}_{k-1}^j - \hat{\mathbf{x}}_{k-1}^j x_{k-2}^{p_j}\right) \right]$

20:      **for** $j \in \mathcal{J}_{\mathcal{Y}}^{(t)}$ **do**

21:          $\widetilde{\mathbf{y}}_k^j = \left[ \hat{\mathbf{y}}_k^j y_{k-1}^{p_j} + \frac{\eta_{k-1}}{\eta_k}\left(\mathbf{y}_{k-1}^j - \hat{\mathbf{y}}_{k-1}^j y_{k-2}^{p_j}\right) \right]$

22:      **for** $j \in \mathcal{J}_{\mathcal{X}}$ **do**

23:          $\mathbf{x}_k^j = x_k^{p_j} \cdot \hat{\mathbf{x}}_k^j$

24:      **for** $j \in \mathcal{J}_{\mathcal{Y}}$ **do**

25:          $\mathbf{y}_k^j = y_k^{p_j} \cdot \hat{\mathbf{y}}_k^j$

26:      $\bar{\mathbf{x}}_k = \frac{H_k - \eta_k}{H_k}\bar{\mathbf{x}}_{k-1} + \frac{\eta_k}{H_k}\mathbf{x}_k, \bar{\mathbf{y}}_k = \frac{H_k - \eta_k}{H_k}\bar{\mathbf{y}}_{k-1} + \frac{\eta_k}{H_k}\mathbf{y}_k$

27: **Return:** $\bar{\mathbf{x}}_K, \bar{\mathbf{y}}_K$

---

## E.2 Liar's Dice

Liar's Dice is another standard benchmark in the EFG-solving community [29]. In the version that we test on, each player rolls an unbiased six-sided die, and they take turns either calling higher bids or challenging the other player. A bid consists of a combination of a value $v$ between one and six, and a number of dice between one and two, $n$, representing the number of dice between the two players that has $v$ pips showing. A higher bid involves either increasing $n$ holding $v$ fixed, increasing $v$ holding $n$ fixed, or both. When a player is challenged (or the highest possible bid of "two dice each showing six pips" is called), the dice are revealed, and whoever is correct wins 1 (either the challenger if the bid is not true, or the player who last called a bid, if the bid is true), and the other player receives a utility of -1.

## E.3 Leduc (13 ranks)

Leduc is yet another standard benchmark in the EFG-solving community [40] and is a simplified version of Texas Hold'Em. In the version we test on, there are 13 unique cards (ranks), and there are 2 copies of each rank (half the size of a standard 52 card deck). There are two rounds of betting that take place, and before the first round each player places an ante of 1 into the pot, and is dealt a single pocket (private) card. In addition, two cards are placed face down, and these are community cards that will be used to form hands. The two hands that can be formed with the community cards are pair, and highest card.

During the first round of betting, player 1 acts first. There is a max of two raises allowed in each round of betting. Each player can either check, raise, or fold. If a player folds, the other player immediately wins the pots and the game terminates. If a player checks, the other player has an opportunity to raise if they have not already previously checked or raised, and if they previously checked, the game moves on to the next round. If a player raises, the other player has an opportunity to raise if they have not already previously raised. The game then moves on the second round, during which one of the community cards is placed face up, and then similar betting dynamics as the first round take place. After the second round terminates, there is a showdown, and whoever can form the better hand (higher ranked pair, or highest card) with the community cards takes the pot.

## E.4 Battleship

This is an instantiation of the classic board game, Battleship, in which players take turns shooting at the opposing player's ships. Before the game begins, the players place two ships of length 2 and value 4, on a grid of size 2 by 3. The ships need to be placed in a way so that the ships take up exactly four spaces within the grid (they do not overlap with each other, and are contained entirely in the grid). Each player gets three shots, and players take turns firing at cells of their opponent's grid. A ship is sunk when the two cells it has been placed on have been shot at. At the end of the game, the utility for a player is the difference between the cumulative value of the opponent's sunk ships and the cumulative value of the player's sunk ships.

# F    Block Construction Strategies

As discussed in the main paper, the postorder block construction strategy can be viewed as traversing the decision points of the treeplex in postorder, treating decision points with the same parent sequence as the same node, and then greedily putting decision points in the same block until we reach a decision point that has a child decision point in the current block (at which point we start a new block). We make this postorder traversal and greedy block construction explicit in Algorithm 3.

In Algorithm 4 we provide pseudocode for constructing blocks using the children block construction strategy. As discussed in the main paper, the children block construction strategy corresponds to placing decision points with the same parent decision point (same decision point at which their parent sequences start at) in the same block. In our implementation, instead of doing a bottom-up traversal, we do a top down implementation, and at the end, reverse the order of the blocks (this allows us to respect the treeplex ordering).

In both Algorithm 3 and Algorithm 4, $\emptyset$ represents the empty sequence.

---

**Algorithm 3** Postorder Block Construction

---

1: **procedure** POSTORDERHELPER($j, a$)
2:     accumulator $= []$
3:     **for** $j' \in \mathcal{C}_{j,a}$ **do**
4:         **for** $a' \in A_{j'}$ **do**
5:             accumulator.insert(postorder($j', a'$))
6:     **for** $j' \in \mathcal{C}_{j,a}$ **do**
7:         accumulator.insert($j'$)
8:     **Return:** accumulator
9: **procedure** POSTORDERBLOCKS($\mathcal{J}$)
10:     ordered = POSTORDERHELPER($\emptyset$)
11:     blocks $= []$
12:     current_block $= []$
13:     **for** $j \in$ ordered **do**
14:         **if** $\exists j' \in$ current_block s.t. $j'$ is a child decision point of $j$ **then**
15:             blocks.insert(current_block)
16:             current_block $= [j]$
17:         **else**
18:             current_block.insert($j$)
19:     **return:** blocks

---

---
**Algorithm 4** Children Block Construction
---
1: **procedure** CHILDRENBLOCKS($\mathcal{J}$)
2:     blocks = []
3:     explore = $\mathcal{C}_\emptyset$
4:     **for** $j \in$ explore **do**
5:         current_block = []
6:         **for** $a \in A_j$ **do**
7:             **for** $j' \in \mathcal{C}_{j,a}$ **do**
8:                 current_block.insert(j')
9:                 explore.insert(j')
            blocks.insert(current_block)
10:     **return:** blocks.reverse()
---

We can now illustrate each of the block construction strategies on the treeplex for player 1 in Kuhn that was presented in Appendix B. If we use single block, then we have $\mathcal{J}_\mathcal{X}^{(1)} = \mathcal{J}_\mathcal{X} = \{0, 1, 2, 3, 4, 5, 6\}$. If we use infosets, then we have $\mathcal{J}_\mathcal{X}^{(i)} = \{7 - i\}$ for $i \in \{1, 2, 3, 4, 5, 6, 7\}$ (we have to subtract in order to label the infosets in a manner that respects the treeplex ordering). If we use children, then we have $\mathcal{J}_\mathcal{X}^{(1)} = \{4\}, \mathcal{J}_\mathcal{X}^{(2)} = \{5\}, \mathcal{J}_\mathcal{X}^{(3)} = \{6\}, \mathcal{J}_\mathcal{X}^{(4)} = \{1, 2, 3\}$, and $\mathcal{J}_\mathcal{X}^{(5)} = \{0\}$. If we use postorder, then we have $\mathcal{J}_\mathcal{X}^{(1)} = \{4, 5, 6\}, \mathcal{J}_\mathcal{X}^{(2)} = \{1, 2, 3\}$, and $\mathcal{J}_\mathcal{X}^{(3)} = \{0\}$.

Note that in the implementation of our algorithm, it is not actually important that the number of blocks for both players are the same; if one player has more blocks than the other, for iterations of our algorithm that correspond to block numbers that do not exist for the other player, we just do not do anything for the other player. Nevertheless, the output of the algorithm does not change if we combine all the blocks for the player with more blocks after the minimum number of blocks between the two players is exceeded, into one block. For example, if player 1 has $s_1$ blocks, and player 2 has $s_2$ blocks, with $s_1 < s_2$, we can actually combine blocks $s_1 + 1, \ldots, s_2$ all into the same block for player 2, and this would not change the execution of the algorithm. This is what we do in our implementation.

Additionally, given a choice of a partition of decision points into blocks, there may exist many permutations of decision points within the blocks which satisfy the treeplex ordering of the decision points. Unless the game that is being tested upon possesses some structure which leads to a single canonical ordering of the decision points (which respects the treeplex ordering), an arbitrary decision needs to be made regarding what order is used.

## G Experiments

### G.1 Additional Experimental Details

**Block Construction Strategy Comparison.** In this section, we provide additional plots (Figures 6 to 14) comparing different block construction strategies for our algorithm, for specific choices of regularizer and averaging scheme. Note that for the games for which there is a benefit to using blocks (Liar's Dice and Battleship), the benefit is generally apparent across different regularizers and averaging schemes. Furthermore, when there is not a benefit for a particular regularizer and averaging scheme, there is no significant cost either (using blocks does not lead to worse performance).

**Block Construction Strategy Comparison with Restarts.** We repeat a similar analysis as above (comparing the block construction strategies holding a regularizer and averaging scheme fixed) but this time with the adaptive restarting heuristic applied to our algorithm: the plots can be seen in (Figures 15 to 23). As mentioned in Section 4, we prepend the algorithm name with "r" to denote the restarted variant of the algorithm in the plots.

As discussed in the main body, the trend of the benefit of using blocks being more pronounced with restarting (for games for which blocks are beneficial) holds generally even when holding the regularizer and averaging scheme fixed. This can be seen by comparing each of the restarted block construction strategy comparison plots with the corresponding non-restarted block construction strategy comparison plot.

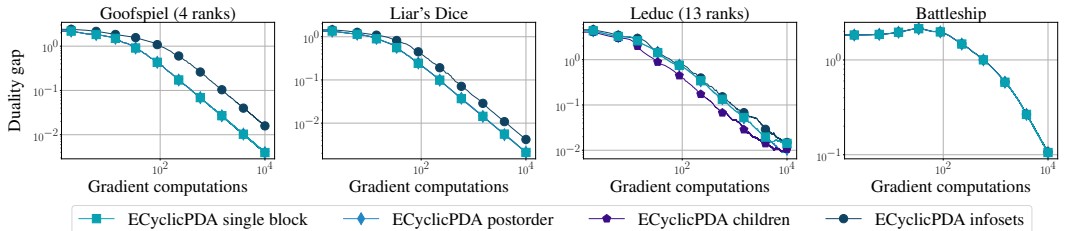

Figure 6: Duality gap as a function of the number of full (or equivalent) gradient computations for ECyclicPDA with different block construction strategies when using the dilated entropy regularizer and uniform averaging.

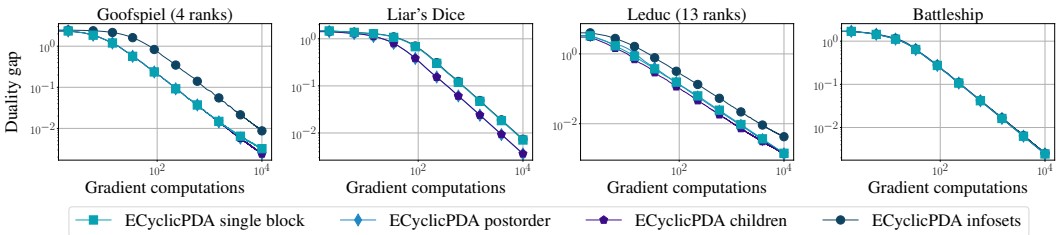

Figure 7: Duality gap as a function of the number of full (or equivalent) gradient computations for ECyclicPDA with different block construction strategies when using the dilatable global entropy regularizer and uniform averaging.

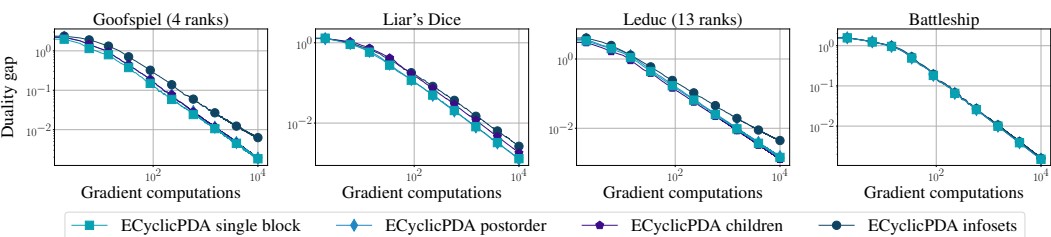

Figure 8: Duality gap as a function of the number of full (or equivalent) gradient computations for ECyclicPDA with different block construction strategies when using the dilated $\ell^2$ regularizer and uniform averaging.

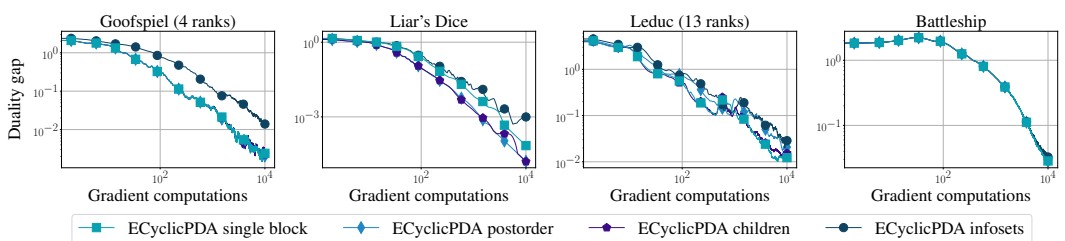

Figure 9: Duality gap as a function of the number of full (or equivalent) gradient computations for ECyclicPDA with different block construction strategies when using the dilated entropy regularizer and linear averaging.

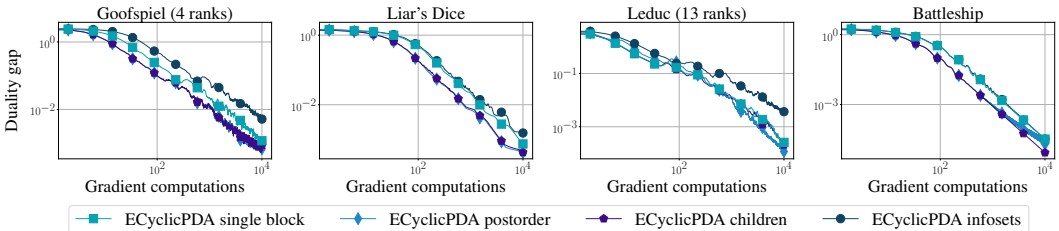

Figure 10: Duality gap as a function of the number of full (or equivalent) gradient computations for ECyclicPDA with different block construction strategies when using the dilatable global entropy regularizer and linear averaging.

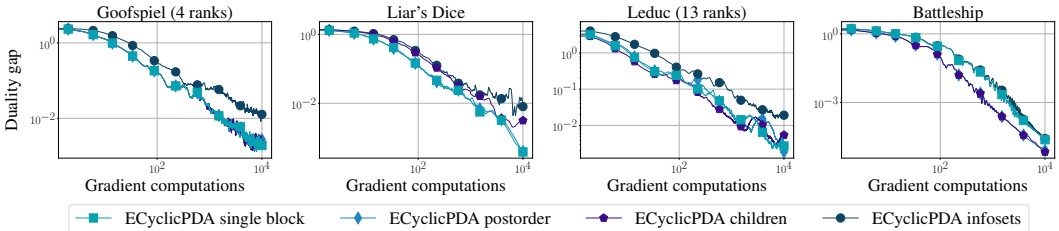

Figure 11: Duality gap as a function of the number of full (or equivalent) gradient computations for ECyclicPDA with different block construction strategies when using the dilated $\ell^2$ regularizer and linear averaging.

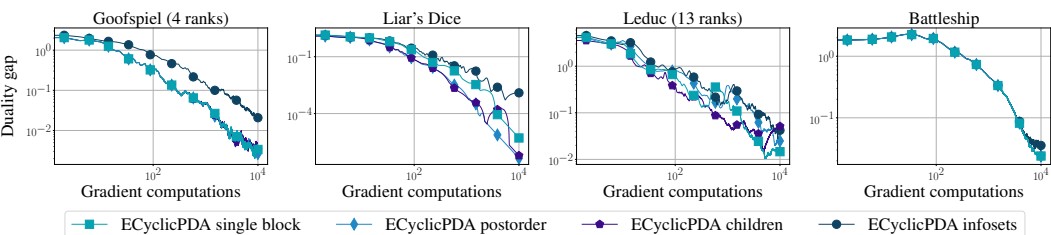

Figure 12: Duality gap as a function of the number of full (or equivalent) gradient computations for ECyclicPDA with different block construction strategies when using the dilated entropy regularizer and quadratic averaging.

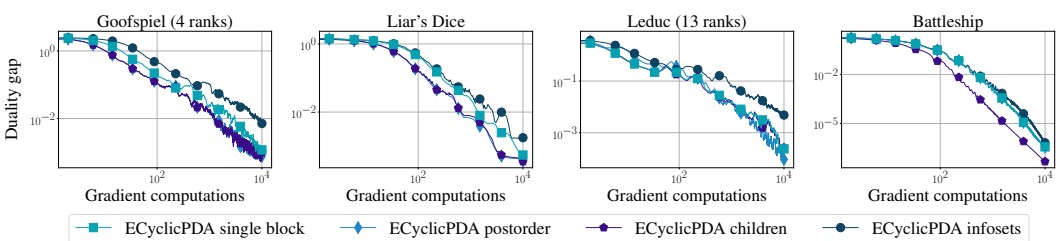

Figure 13: Duality gap as a function of the number of full (or equivalent) gradient computations for ECyclicPDA with different block construction strategies when using the dilatable global entropy regularizer and quadratic averaging.

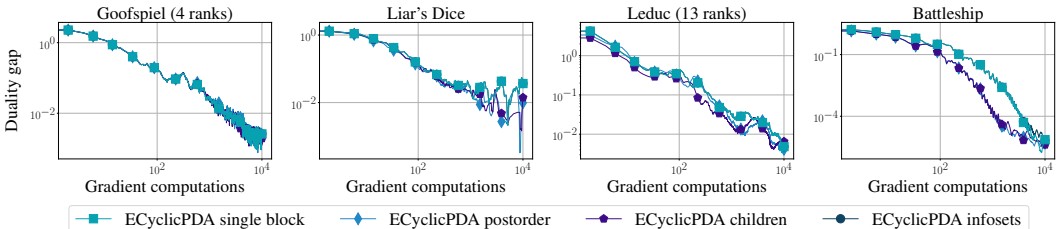

Figure 14: Duality gap as a function of the number of full (or equivalent) gradient computations for ECyclicPDA with different block construction strategies when using the dilated $\ell^2$ regularizer and quadratic averaging.

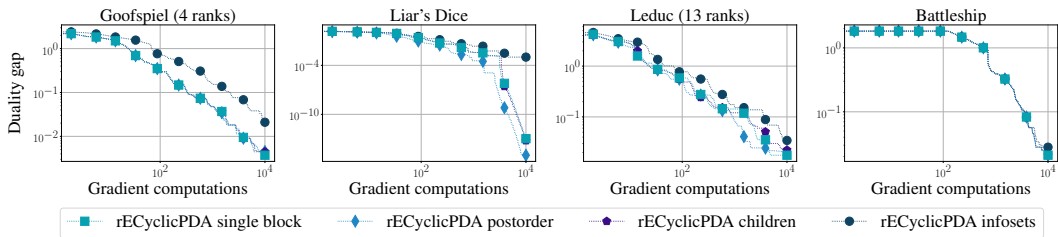

Figure 15: Duality gap as a function of the number of full (or equivalent) gradient computations for ECyclicPDA with different block construction strategies when using the dilated entropy regularizer and uniform averaging as well as restarting. We take the best duality gap seen so far so that the plot is monotonic.

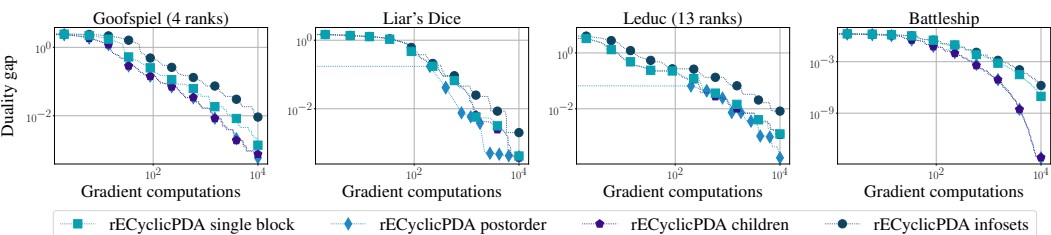

Figure 16: Duality gap as a function of the number of full (or equivalent) gradient computations for ECyclicPDA with different block construction strategies when using the dilatable global entropy regularizer and uniform averaging as well as restarting. We take the best duality gap seen so far so that the plot is monotonic.

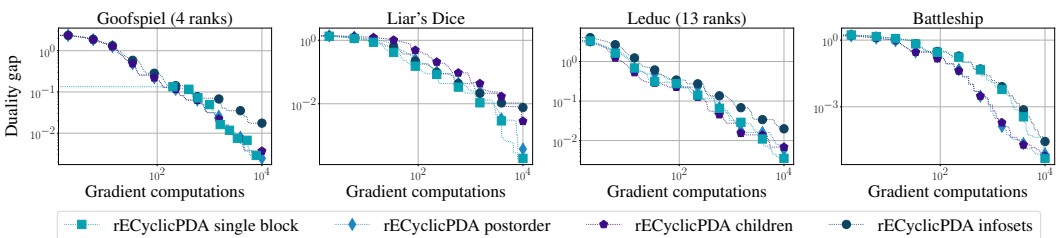

Figure 17: Duality gap as a function of the number of full (or equivalent) gradient computations for ECyclicPDA with different block construction strategies when using the dilated $\ell^2$ regularizer and uniform averaging as well as restarting. We take the best duality gap seen so far so that the plot is monotonic.

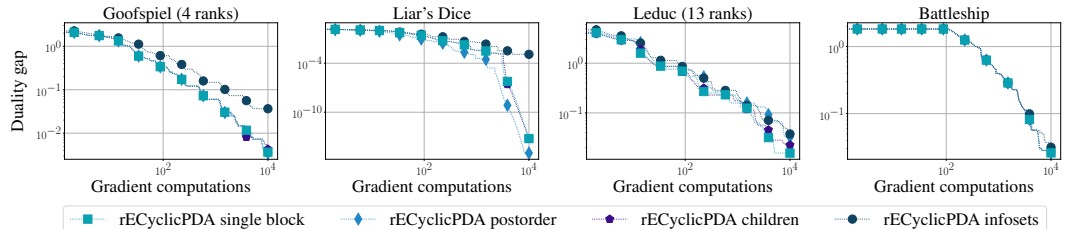

Figure 18: Duality gap as a function of the number of full (or equivalent) gradient computations for ECyclicPDA with different block construction strategies when using the dilated entropy regularizer and linear averaging as well as restarting. We take the best duality gap seen so far so that the plot is monotonic.

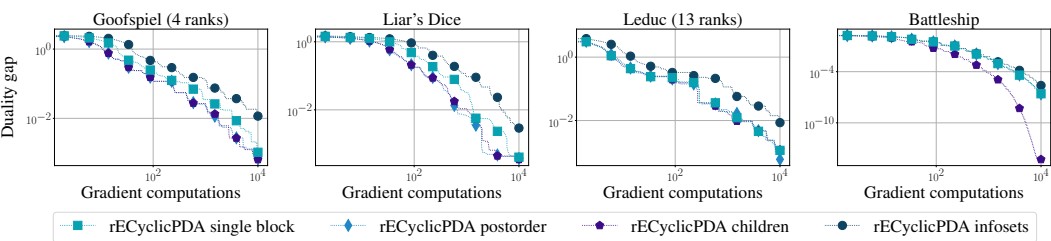

Figure 19: Duality gap as a function of the number of full (or equivalent) gradient computations for ECyclicPDA with different block construction strategies when using the dilatable global entropy regularizer and linear averaging as well as restarting. We take the best duality gap seen so far so that the plot is monotonic.

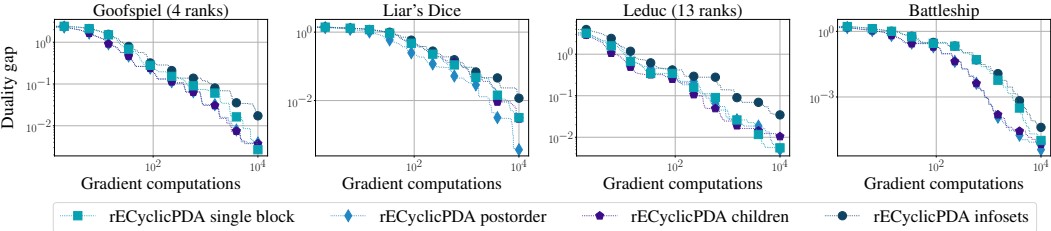

Figure 20: Duality gap as a function of the number of full (or equivalent) gradient computations for ECyclicPDA with different block construction strategies when using the dilated $\ell^2$ regularizer and linear averaging as well as restarting. We take the best duality gap seen so far so that the plot is monotonic.

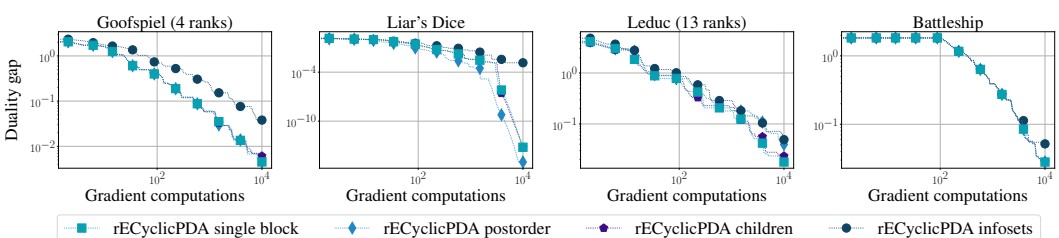

Figure 21: Duality gap as a function of the number of full (or equivalent) gradient computations for ECyclicPDA with different block construction strategies when using the dilated entropy regularizer and quadratic averaging as well as restarting. We take the best duality gap seen so far so that the plot is monotonic.

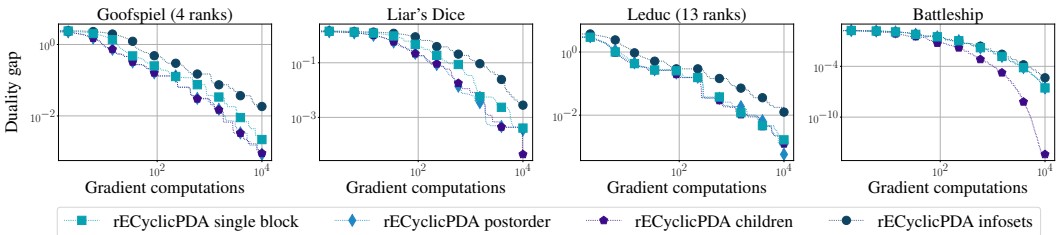

Figure 22: Duality gap as a function of the number of full (or equivalent) gradient computations for ECyclicPDA with different block construction strategies when using the dilatable global entropy regularizer and quadratic averaging as well as restarting. We take the best duality gap seen so far so that the plot is monotonic.

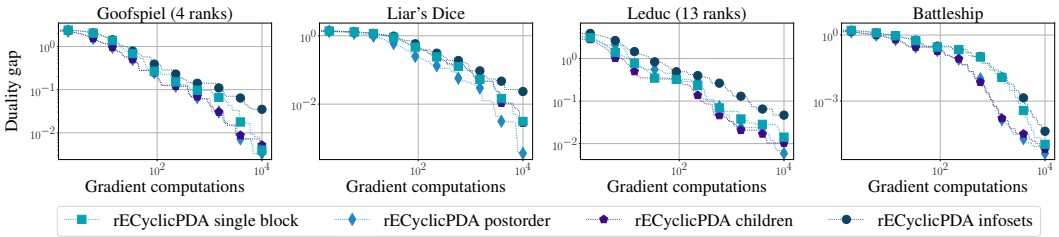

Figure 23: Duality gap as a function of the number of full (or equivalent) gradient computations for ECyclicPDA with different block construction strategies when using the dilated $\ell^2$ regularizer and quadratic averaging as well as restarting. We take the best duality gap seen so far so that the plot is monotonic.

**Regularizer Comparison.** In this section (Figures 24 to 26) we compare the performance of ECyclicPDA and MP instantiated with different regularizers for each averaging scheme, against the performance of CFR$^+$ and PCFR$^+$.

It is apparent from these plots, that our algorithm generally outperforms MP, holding the averaging scheme and regularizer fixed. This can be seen by examining the corresponding figure for a choice of averaging scheme, and noting that for any given regularizer, the corresponding MP line is generally above the corresponding ECyclicPDA line.

**Regularizer Comparisons with Restarts.** We repeat a similar analysis in this section (Figures 27 to 29), instead now comparing the performance of ECyclicPDA and MP instantiated with different regularizers for each averaging scheme, against the performance of CFR$^+$ and PCFR$^+$, when all methods are restarted. The trend noted above of our method generally beating MP, even holding the regularizer and averaging scheme fixed, still holds even when restarting.

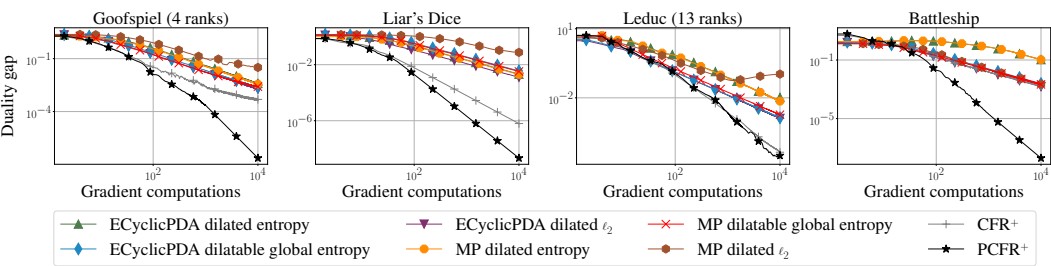

Figure 24: Duality gap as a function of the number of full (or equivalent) gradient computations for ECyclicPDA, MP, CFR$^+$, PCFR$^+$, using a uniform averaging scheme for ECyclicPDA and MP.

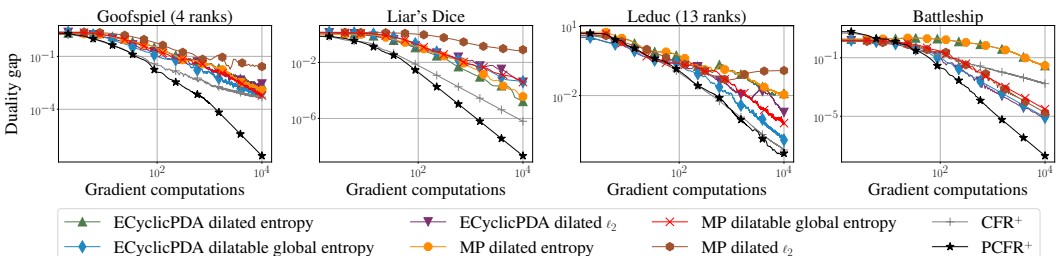

Figure 25: Duality gap as a function of the number of full (or equivalent) gradient computations for ECyclicPDA, MP, CFR$^+$, PCFR$^+$, using a linear averaging scheme for ECyclicPDA and MP.

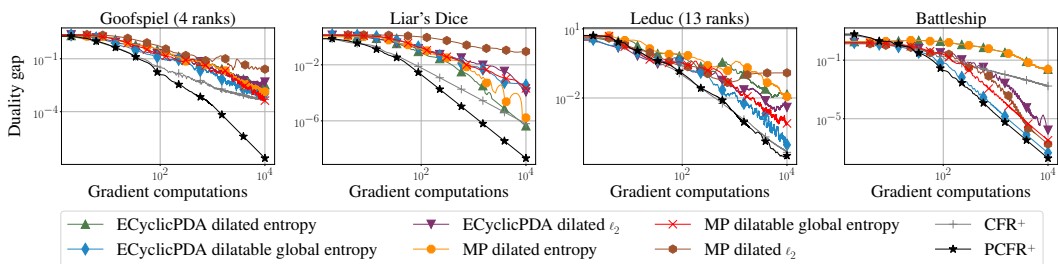

Figure 26: Duality gap as a function of the number of full (or equivalent) gradient computations for ECyclicPDA, MP, CFR$^+$, PCFR$^+$, using a quadratic averaging scheme for ECyclicPDA and MP.

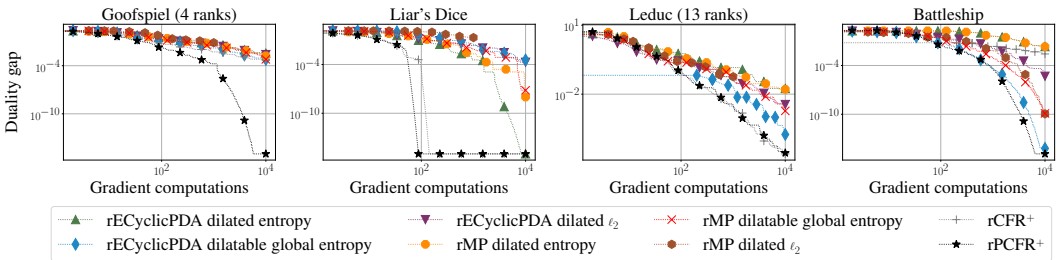

Figure 27: Duality gap as a function of the number of full (or equivalent) gradient computations for when restarting is applied to ECyclicPDA, MP, CFR$^+$, PCFR$^+$, using a uniform averaging scheme for ECyclicPDA and MP. We take the best duality gap seen so far so that the plot is monotonic.

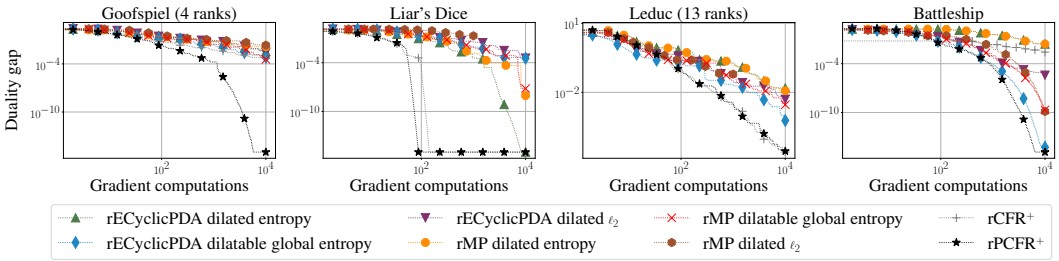

Figure 28: Duality gap as a function of the number of full (or equivalent) gradient computations for when restarting is applied to ECyclicPDA, MP, CFR$^+$, PCFR$^+$, using a linear averaging scheme for ECyclicPDA and MP. We take the best duality gap seen so far so that the plot is monotonic.

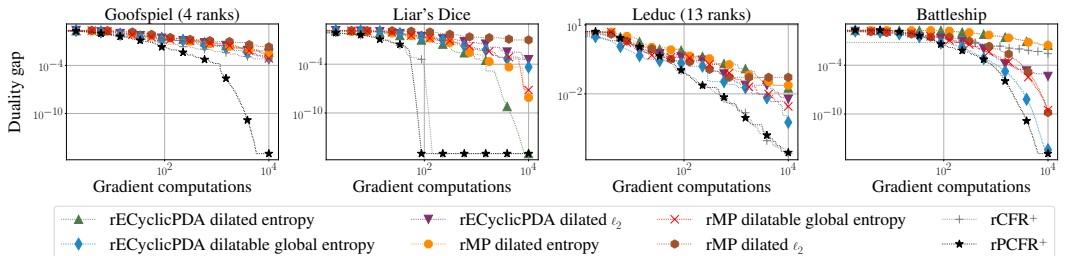

Figure 29: Duality gap as a function of the number of full (or equivalent) gradient computations for when restarting is applied to ECyclicPDA, MP, CFR$^+$, PCFR$^+$, using a quadratic averaging scheme for ECyclicPDA and MP. We take the best duality gap seen so far so that the plot is monotonic.

**Additional wall-clock time experiments** In Table 3, we show the wall-clock time required to reach a duality gap of $10^{-4}$. It is clear that our algorithm is competitive with MP: MP and its restarted variant time out on Leduc, and MP times out on Battleship (while our algorithm does not even take close to 30 seconds). Furthermore, we are outperforming CFR$^+$ and its restarted variant in Battleship.

Table 3: The wall clock time in seconds required for ECyclicPDA, MP, CFR$^+$, and PCFR$^+$, and their restarted variants, denoted using an "r" at the front of the algorithm name, to reach a duality gap of $10^{-4}$. We let each algorithm run for at most 30 seconds; a value of 30.000 means the algorithm could not reach the target gap in 30 seconds. The duality gap is computed every 100 iterations.

| Name | Goofspiel (4 ranks) | Liar's Dice | Leduc (13 ranks) | Battleship |
|---|---|---|---|---|
| ECyclicPDA | 8.797 | 6.148 | 9.343 | 16.818 |
| rECyclicPDA | 6.688 | 3.519 | 10.724 | 4.914 |
| MP | 3.183 | 3.930 | 30.000 | 30.000 |
| rMP | 2.691 | 2.360 | 30.000 | 8.374 |
| CFR$^+$ | 5.869 | 0.111 | 0.236 | 6.001 |
| rCFR$^+$ | 5.866 | 0.057 | 0.161 | 6.042 |
| PCFR$^+$ | 0.244 | 0.067 | 0.225 | 0.736 |
| rPCFR$^+$ | 0.208 | 0.077 | 0.197 | 0.736 |

