# OpenReview forum: "Block-Coordinate Methods and Restarting for Solving Extensive-Form Games"
_NeurIPS.cc/2023/Conference — NeurIPS 2023 poster_

### Official Review · Reviewer_XCKi · 2023-06-27

**Soundness:** 2 fair
**Presentation:** 2 fair
**Contribution:** 2 fair
**Rating:** 5
**Confidence:** 1

**Summary:**

This work proposes a cyclic coordinate descent method to solve the two-player zero-sum extended form game (EFG) and derives the convergence.

**Strengths:**

To me, solving problems with non-separable constraints by coordinate-descent-type methods is novel and interesting. Therefore, I believe that this is indeed a contribution.

**Weaknesses:**

I'm not familiar with this area so please forgave me if I made some mistakes.

1. Possibly expensive computational cost per iteration. Following the literature survey in the manuscript, I read reference [1] which studies coordinate descent for solving optimization problems with non-separable non-smooth objective functions. I found that in both the algorithm in [1] and the algorithm in this paper, although only partial gradient is needed at each update, the proximal step with respect to the whole nonsmooth function is needed. More specifically, in Lines 8 and 11, the argmin step might be difficult and computationally expensive since the argmin step is over the whole X and Y and involve all the blocks.

2.  This concern follows the first one but focuses on the experiments: the comparison with non-coordinate algorithms may be unfair. The current comparison is in terms of the number of full gradient computations. However, gradient computation is not the only computation cost, Lines 8 and 11 also cause computation costs. I would suggest a comparison in terms of wall clock time.



**Questions:**

no

**Limitations:**

I do not see any potential negative societal impact.

---

> ### Author Rebuttal · Authors · 2023-08-10
>
> Thank you for taking the time to review our paper. We provide responses to your concerns below:
>
> > Possibly expensive computational cost per iteration. Following the literature survey in the manuscript, I read reference [1] which studies coordinate descent for solving optimization problems with non-separable non-smooth objective functions. I found that in both the algorithm in [1] and the algorithm in this paper, although only partial gradient is needed at each update, the proximal step with respect to the whole nonsmooth function is needed. More specifically, in Lines 8 and 11, the argmin step might be difficult and computationally expensive since the argmin step is over the whole X and Y and involve all the blocks.
>
> We have addressed this in the top-level comment (5). While the argmin in Lines 8 and 11 of Algorithm 1 might make it seem like we are doing a proximal update with respect to the entire feasible set, in fact, we can do this computation without having to consider the entire feasible set (and this is the primary premise of our paper). Please also see the implementation version of the algorithm in Appendix D that makes this more transparent.
>
> As mentioned in the top-level comment (1-4), the primary contribution of our paper is to provide a coordinate-descent-like method which circumvents the issue of non-separability without having to compute a proximal update with respect to the entire feasible set every time a partial gradient is taken with respect to a block. We do this by combining the recursive prox update structure that dilated regularizers are known  to have (Proposition 2.2) with an extrapolated cyclic method.  We have an implementation-specific algorithm in Appendix D which demonstrates that the computation done for gradient updates and prox updates (which are the computational bottleneck when applying first-order methods to EFGs) for our method is comparable to the respective computations done for MP. In fact, after performing one whole iteration of the outer loop (line 2 in Alg 1), we will have spent almost exactly the same amount of time as a single prox computation + gradient computation when running Mirror Prox. However, we agree that the body could have more clearly explained this, and we will update the paper to reflect this.
>
> > This concern follows the first one but focuses on the experiments: the comparison with non-coordinate algorithms may be unfair. The current comparison is in terms of the number of full gradient computations. However, gradient computation is not the only computation cost, Lines 8 and 11 also cause computation costs. I would suggest a comparison in terms of wall clock time.
>
> We have addressed this in the top-level comment (5) and also earlier in our response. We demonstrate in Appendix D that Lines 8 and 11 (the prox updates for each of the players) require comparable computation to the prox updates done in MP. We will make this more explicit in the main body (as we note in the top-level comment). Note that the wall-clock times of our algorithm are comparable to MP per “full” gradient computation (as shown in Tables 1 and 2 in the attached pdf).

---

> > ### Comment · Reviewer_XCKi · 2023-08-12
> >
> > Thank you for your response.

---

### Official Review · Reviewer_rJqj · 2023-07-05

**Soundness:** 3 good
**Presentation:** 3 good
**Contribution:** 3 good
**Rating:** 6
**Confidence:** 3

**Summary:**

This paper combines the local prox update technique with the extrapolated cyclic algorithm and proposes the ECyclicPDA algorithm. While the local prox update technique is well understood, the authors reinterpret it as a coordinate method (CM). Then, the method is combined with a new extrapolated method. Theoretical analysis shows that the proposed algorithm can converge at a rate of O(1/T).

**Strengths:**

1. This paper presents a new understanding of the local update rules of OMD with dilated DGF. The combination of local OMD updates and extrapolated updates is original and interesting.
2. The theoretical results seem sound.
3. The paper is well-written.

**Weaknesses:**

1. It is hard to understand why we want to reinterpret the well-known local update rules of OMD with dilated DGF to CM. As we know, people already compute the strategy in a bottom-up fashion in previous work [1, 2, 3]. So, I think the main contribution is the combination of the local update method and the extrapolated cyclic method.
2. However, the ECyclicPDA algorithm seems less efficient than the traditional MP algorithm. For every infoset, an extra traversal of the subgame rooted at the infoset is needed to compute the “extrapolated” vector. This could be infeasible for large-scale games.
3. The experimental results show that the proposed ECyclicPDA performs better than MP. The results are not surprising when considering that ECyclicPDA can be very time-consuming.
4. The algorithm does not compare with related optimistic OMD [4] algorithms for EFGs.


[1] Farina, Gabriele, Christian Kroer, and Tuomas Sandholm. "Optimistic regret minimization for extensive-form games via dilated distance-generating functions." Advances in neural information processing systems 32 (2019).
[2] Farina, Gabriele, Christian Kroer, and Tuomas Sandholm. "Better regularization for sequential decision spaces: Fast convergence rates for Nash, correlated, and team equilibria." arXiv preprint arXiv:2105.12954 (2021).
[3] Liu, Weiming, et al. "Equivalence analysis between counterfactual regret minimization and online mirror descent." International Conference on Machine Learning. PMLR, 2022.
[4] Lee, Chung-Wei, Christian Kroer, and Haipeng Luo. "Last-iterate convergence in extensive-form games." Advances in Neural Information Processing Systems 34 (2021): 14293-14305.

**Questions:**

none

**Limitations:**

The authors have stated the limitations properly. No potential negative societal impact.

---

> ### Author Rebuttal · Authors · 2023-08-10
>
> Thank you for taking the time to review our paper. We provide responses to your concerns below:
>
> > It is hard to understand why we want to reinterpret the well-known local update rules of OMD with dilated DGF to CM. As we know, people already compute the strategy in a bottom-up fashion in previous work [1, 2, 3]. So, I think the main contribution is the combination of the local update method and the extrapolated cyclic method.
>
> We mention in our paper and in the top-level comment (3), that the recursive computation of the prox update when using dilated regularizers in EFGs is well-known. The main contribution of *our* paper is different though (as mentioned in top-level comment (4)): what we are showing is that the ``local update’’ structure of dilated DGFs enables us to perform extrapolated cyclic block-coordinate updates. This is completely new, and different from the purpose of the local updates in prior works. In prior works the local updates were merely a way to efficiently implement deterministic full-gradient updates.
>
> > However, the ECyclicPDA algorithm seems less efficient than the traditional MP algorithm. For every infoset, an extra traversal of the subgame rooted at the infoset is needed to compute the “extrapolated” vector. This could be infeasible for large-scale games
>
> We have discussed this issue in the top level comment (5) and there is an analysis of the per-iteration complexity provided in Appendix D, where we note that the per-iteration complexity is comparable to that of MP. The scaled values used for extrapolation can be computed while we are traversing the treeplex to do the partial prox update; note that because of the recursive nature of the prox update (as discussed in the top level comment (3,5)), this does not require any extra traversal through the treeplex as compared to a full-gradient method and thus neither does the computation of the “extrapolated vector”. We will make this more explicit in the main body in a revised version of the paper.
>
> > The experimental results show that the proposed ECyclicPDA performs better than MP. The results are not surprising when considering that ECyclicPDA can be very time-consuming.
>
> We have discussed this in our top level comment (5), and the attached pdf contains data on the runtime of our algorithm and the other algorithms we compare against. ECyclicPDA does not take more time than MP (in fact, it usually takes *less* time per outer iteration) and performs about half as many arithmetic operations per outer iteration (note that one outer iteration for ECyclicPDA corresponds to a full gradient update, and is thus is comparable to one iteration of MP).
>
> > The algorithm does not compare with related optimistic OMD [4] algorithms for EFGs.
>
> We will add this in a revised version of the paper; we were not able to obtain results for this in time for the rebuttal. We suspect that the performance will be comparable to MP, perhaps very slightly better, based on prior work.

---

> > ### Comment · Reviewer_rJqj · 2023-08-13
> >
> > Thank you for your response. My confusion has been completely resolved, and therefore I will raise my rating.

---

### Official Review · Reviewer_Y4SP · 2023-07-09

**Soundness:** 3 good
**Presentation:** 3 good
**Contribution:** 3 good
**Rating:** 7
**Confidence:** 3

**Summary:**

This paper introduces ECyclicPDA, a new first order method for solving extensive form games.  The main idea is to implement something in the spirit of coordinate descent where improving directions can be found by considering a part of the current iterate in isolation.  Empirical results show it generally outperforms other first-order methods and approaches the performance of CFR+.  Additionally, a heuristic to restart the process is introduced which improves the performance of both ECyclicPDA and CFR+

**Strengths:**

ECyclicPDA is a nice contribution that takes the growing collection of FOMs for EFGs in a new direction.  The analysis seems to involve non-trivial technical innovations.  The empirical results are convining and show continued progress toward cloasing the gap with CFR-based approaches.

The restarting heuristic is also very interesting, and the ability to get performance improvements with CFR+ is particularly nice (and intuitive in hindsight).


**Weaknesses:**

As far as I can tell, the details of how restarting is implemented are never clearly explained.  The clearest explanation I can find is in the introduction on lines 88-91, but even taking that as the full specification, I don’t see where the tuning of the parameter about when to restart is specified.  It would also be nice to have a bit of explanation / intuition for how restarting is benefiting various algorithms.  For CFR+ I can see how resetting the regret sums and the averaging process could be beneficial.  For FOMs I’m less clear why it is useful.  Is it purely from resetting the averaging process?

**Questions:**

Please comment on the questions about resetting.

**Limitations:**

Adequate

---

> ### Author Rebuttal · Authors · 2023-08-10
>
> Thank you for taking the time to review our paper. We are glad that you found our contributions interesting. We provide responses to your concerns below:
>
> > As far as I can tell, the details of how restarting is implemented are never clearly explained. The clearest explanation I can find is in the introduction on lines 88-91, but even taking that as the full specification, I don’t see where the tuning of the parameter about when to restart is specified. It would also be nice to have a bit of explanation / intuition for how restarting is benefiting various algorithms. For CFR+ I can see how resetting the regret sums and the averaging process could be beneficial. For FOMs I’m less clear why it is useful. Is it purely from resetting the averaging process?
>
> In our numerical experiments, we are restarting the algorithms every time the duality gap has been halved and initializing the next run at the output (averaged) iterate. We did not explore tuning parameters pertaining to when to restart. We will update this in a revision of our paper to make clear how restarting is implemented in our experiments. For FOMs, we believe that the primary benefit comes from resetting the averaging process, though it is somewhat unclear whether there exists a more intuitive or deeper explanation than this.

---

> > ### Comment · Reviewer_Y4SP · 2023-08-11
> >
> > Thank you for the response

---

### Official Review · Reviewer_udTT · 2023-07-09

**Soundness:** 3 good
**Presentation:** 4 excellent
**Contribution:** 2 fair
**Rating:** 6
**Confidence:** 3

**Summary:**

The draft considers solving the extended-form game (EFG) with extrapolated block-coordinate descent methods, proving that it achieves O(1/T) convergence. The authors further show that with a restarting strategy, the proposed algorithm may be comparable sometimes to state-of-the-art algorithms like CFR+. In my understanding, the proposed algorithm is essentially a generalization of the CODER algorithm proposed by Song [37] in a way from coordinate descent (CD) to blockwise CD and specializes the domain to be linear functional and treeplex. I.e., the algorithm is essentially the same (line-by-line corresponded) as CODER but generalizes the scalar coordinate to blocks of "separable" variables.  For the specialization, the authors focus on the context of EFG with bilinear score function and 2-player treeplex (linear probability simplex), where the prox is taken on L1-norm instead of the L2-norm by CODER. Implementation-wise, the proposed algorithm uses the recursive structure of the treeplex for efficient linear time computation.

**Strengths:**

The major contribution of the draft is that it provides an instance to efficiently implement the CODER algorithm on EFG with blockwise L1 proximal and shows that with restarting, such implementation may be comparable with the state-of-the-art algorithm like CFR+. It also analyzes the scenario and provides an O(1/T) convergence rate without dependence on the number of variables. To summarize, the draft shows that a simple CODER extrapolation works well when taking L1 specialization on EFG.

**Weaknesses:**

However, I would like to claim that the original CODER method may already cover the blockwise updates. I.e., if you look into the proof of CODER, it generally does not use the scalar property of the proximal and treats variables as blocks. They are using the $d$-dimension notation for simplicity and already claiming they are doing block CD in their abstract. In the EFG case, the d=2, so naturally (if the proximal works), the convergence complexity doesn't depend on the number of variables since blocks=2 is a constant. So the draft's theoretical result (independence on the number of variables) is not surprising. And the proposed method does not relax the "separability assumption" since the 2-player treeplex is separable as two blocks. The major difference/contribution here is that it performs the analysis on the L1 proximal and does the analysis.

Further, although it's good to see that the simple method works with restarting, the restarting part doesn't have theoretical support. I.e., without restarting, the theoretically supported method is not comparable to the state-of-the-art. And with restarting, the "full" method is comparable but not really much better than CFR+. Thus these factors put the paper in a borderline condition. However, the analysis seems solid, and I would give a borderline acceptance.

**Questions:**

1. (Figure 2) Only doing the step-size optimization for your method and not the baseline is tricky for the experiment. For fairness, it should be either all constant step-size or all step-size tuned for the baseline. Otherwise, the reader cannot be sure whether the proposed method is better than MP.
2. (Figure 4) The vertical drop in the curves looks like either a numerical issue or a wrong optimal value.
3. (Algorithm 1) Adding parenthesis and denoting what you maintain in the computation may be good for clarity.

**Limitations:**

Not applicable since the paper does not have a limitation section.

---

> ### Author Rebuttal · Authors · 2023-08-10
>
> > However, I would like to claim that the original CODER method may already cover the blockwise updates. [...]  So the draft's theoretical result (independence on the number of variables) is not surprising. And the proposed method does not relax the "separability assumption" since the 2-player treeplex is separable as two blocks. The major difference/contribution here is that it performs the analysis on the L1 proximal and does the analysis.
>
> We would like to clarify a few points here that may have been missed by the reviewer. We will make sure to highlight them more in the revision.
>
> First, CODER is not directly applicable to our setting and it is not only because of ell_1 vs ell_2 settings. As discussed in the second paragraph of “Contributions” in the introduction and in the introductory part of Section 3, and in our top-level comment (1), our feasible region (treeplex) is not block separable (except for the trivial primal-dual separation mentioned in the review), hence it violates one of the main assumptions made in CODER. Note that block coordinate algorithms are mainly useful and applied with many small blocks (see, e.g., the block partition strategies described in our experimental section). Even ignoring this point and trying to apply CODER bottom up, there are at least two other issues. First, the extrapolation step in CODER would require exact (non-scaled) values of the blocks of $\mathbf{x}_k$ and $\mathbf{y}_k$ that got traversed up to the cycle iteration $t$, which are simply not available without a full tree traversal – these values are only available up to normalizing factors that can get computed only at the end of the bottom-up traversal, in the top-down (rescaling) traversal (Lines 13-16 in Algorithm 1). This fundamentally changes the analysis. Second, we do a block partition on both the primal and the dual side, thus the number of blocks is $2 \cdot m$ (not $2$, as there are $m$ blocks per player and the number of blocks can be as large as the number of infosets, which can be order-dimension). Even if CODER’s analysis was applicable here, its convergence bound would scale with $\sqrt{m}$. Our analysis does not incur such a dependence and this comes precisely from a careful interleaving argument carried out in the proof of Theorem 3.2 (please see Lemma C.1 and its proof in the appendix).
>
> > Further, although it's good to see that the simple method works with restarting, the restarting part doesn't have theoretical support. I.e., without restarting, the theoretically supported method is not comparable to the state-of-the-art. And with restarting, the "full" method is comparable but not really much better than CFR+. Thus these factors put the paper in a borderline condition. However, the analysis seems solid, and I would give a borderline acceptance.
>
> As we have discussed in our top level comment (6, 7), the practical performance of CFR+ and its variants is not well understood, and not within the scope of our paper (also note that the theoretical guarantees for CFR+ and its variants are worse than the guarantees that exist for MP as well as our method). Our primary contribution is theoretical, and it is surprising that a first-order method is able to compete with it. Furthermore, experimentally, it is a contribution of our paper to demonstrate that “restarting” works well even as a heuristic for our method, MP, and the CFR variants. To the best of our knowledge, this has not been discussed in the literature previously.
>
> > (Figure 2) Only doing the step-size optimization for your method and not the baseline is tricky for the experiment. For fairness, it should be either all constant step-size or all step-size tuned for the baseline. Otherwise, the reader cannot be sure whether the proposed method is better than MP.
>
> Step size optimization is done for MP (as noted in lines 293-294) as well as our method.
> > (Figure 4) The vertical drop in the curves looks like either a numerical issue or a wrong optimal value.
>
> The vertical drop in the curves is not a numerical issue since precision on the order of $10^{-15}$ can be reasonably computed and represented as a double. The algorithms appear to exhibit a linear rate-like convergence when they have that sudden drop-off, which is actually a feature of using the restarting heuristic; this demonstrates that our restarting heuristic works well.
> > (Algorithm 1) Adding parenthesis and denoting what you maintain in the computation may be good for clarity.
>
> As we mention in the top-level comment (5), we have an algorithmic specific implementation in Appendix D that makes clear what quantities are maintained in the computation. The algorithm presented in the main body is for convenience of presenting the main ideas without a too detailed exposition, and for theoretical analysis.

---

> > ### Comment · Reviewer_udTT · 2023-08-16
> >
> > > First, CODER is not directly applicable
> >
> > CODER is not directly applicable to every leaf node, but it is applicable with $d=2$ as I mentioned, that is, separating the variables into only 2 large blocks (two players). The proof in CODER does apply to variable blocks. And I understand your contribution on the L1 norm proof is different form CODER's L2 proof. Just highlighting the relationship of the L2 to L1 generalization, which also appears in other proximal papers.
> >
> > After reading the authors' rebuttal, I decided to keep my rating.

---

> > > ### Author Response · Authors · 2023-08-17
> > >
> > > We would like to thank you for continuing to engage in a discussion about our paper.
> > >
> > > Yes, CODER is applicable only in the trivial cases as we acknowledged in our response (see "except for the trivial primal-dual separation mentioned in the review" in the response above), but not in the general block coordinate case that we focus on in our paper. This is explained in detail in the above response and in the paper. In particular, it is inapplicable for three of the four block construction strategies (children, postorder, and infosets) that we discuss in Section 4. Note that what we are doing is more general than treating “the leaf nodes” as blocks.
> > >
> > > The fourth block construction strategy we consider consists of using a single block for each player and coincides with the trivial decomposition you mention, and thus is already covered by our experiments. Note that this latter trivial block construction strategy corresponds to the well-known scheme of *alternation* in a two-player zero-sum game. Alternation is already known to work e.g., in the context of self-play via regret minimization, as well as in e.g., the Chambolle & Pock primal-dual algorithm. Alternation is quite special, in that it leverages the primal-dual structure of a zero-sum game, and we do not think it is meaningfully a form of “block decomposition” in the spirit of what our paper is accomplishing. Note also that existing work does not describe this as a form of block decomposition.

---

### Official Review · Reviewer_u8q2 · 2023-07-13

**Soundness:** 3 good
**Presentation:** 3 good
**Contribution:** 2 fair
**Rating:** 5
**Confidence:** 1

**Summary:**

The paper introduces a novel method for solving Extensive-Form Games based on a block-coordinate approach. The authors motivate and explain their idea and experimentally evaluate its performance in terms of primal-dual gap in four different games. The method features a favorable theoretical convergence rate but the empirical results are often comparable or worse than existing methods in terms of the duality gap.

**Strengths:**

- The paper is well written and free of grammar and stylistic issues.
- The introduction to the problem is very approachable even to domain non-experts.
- Experimental validation of the block construction strategy choice.
- Favorable converge properties.

**Weaknesses:**

- It seems that in empirical studies the method is consistently worse than PCFR+ (Fig 2 and 4). I have a difficulty to find a complete justification for this in the paper.
- Only a theoretical convergence rate and not an actual computation cost measured in units of time is reported.
- The plots lack error bars (unless no variance is possible to obtain).
- The contributions are explained indirectly. They could be listed in a more compact and more explicit form (e.g., a numbered list) to allow for critical assessment.

**Questions:**

- How does the run-time of the tested methods compare in practice?

*** Rebuttal Acknowledgment ***

I have read the author's rebuttal. The answer covers my questions well and the explanation the authors provides logical justification of the potential shortcomings. Therefore, I am of the opinion that the paper is sound and I increased my rating accordingly. I do not go higher due to my very limited familiarity with the subfield.

**Limitations:**

- The discussion of limitation is not very thorough and it is mostly limited to a mention of a lack of understanding of the game type effect on the performance.

---

> ### Author Rebuttal · Authors · 2023-08-10
>
> Thank you for taking the time to review our paper. We provide responses to your concerns below:
>
> > It seems that in empirical studies the method is consistently worse than PCFR$^+$ (Fig 2 and 4). I have a difficulty to find a complete justification for this in the paper.
>
> As we have discussed in our top level comment (6, 7), the practical performance of CFR$^+$ and its variants is not well understood, and not within the scope of our paper. Our primary contribution is theoretical, and it is surprising that a first-order method is able to compete with it. Moreover, as stated in the overall comment, "PCFR$^+$"  in Fig 4 is actually restarted PCFR$^+$, a new method introduced by us, and we have made this clear in a revision of our paper (including using "rPCFR$^+$"  to denote this adaptive restarting variant that we introduce).
>
> > Only a theoretical convergence rate and not an actual computation cost measured in units of time is reported.
>
> We discuss this in our top-level comment (5). There is a runtime analysis that demonstrates that our per-iteration complexity is comparable to that of Mirror Prox, and we provide tables reporting results in units of time.
>
> > The plots lack error bars (unless no variance is possible to obtain).
>
> Our method and all methods compared against are deterministic so there are no relevant error bars to report.
>
> > How does the run-time of the tested methods compare in practice?
>
> We have provided wall-clock comparisons in the top-level comment (5).
>
> > The discussion of limitation is not very thorough and it is mostly limited to a mention of a lack of understanding of the game type effect on the performance.
>
> We appreciate the suggestion. In a revised version, we will create an explicit limitations section and add discussion of comparisons to CFR$^+$ variants, as well as note that our algorithm performs better on larger games.

---

> > ### Comment · Reviewer_u8q2 · 2023-08-11
> >
> > Thank you for the response.

---

### Official Review · Reviewer_Lrca · 2023-07-24

**Soundness:** 3 good
**Presentation:** 3 good
**Contribution:** 3 good
**Rating:** 7
**Confidence:** 1

**Summary:**


This paper develops a cyclic block-coordinate-descent-like method for two-player zero-sum extensive-form games (EFG). Such methods for EFG are  difficult due to non-separable nature of block structure of the problem. The decision problem for a player in a EFG can be formulated using Treeplex, for which regularizing functions can be constructed through the framework of dilated regularizers. These dilated regularizing functions allow recursive prox computations. This paper utilizes this frame work to develop an extrapolated cyclic algorithm to perform pseudo-block updates.  They demonstrate O(1/T) convergence rate for two-player zero-sum Nash equilibrium  using this method, and provide a specific algorithmic implementation which shows that runtime of the proposed method  is independent of the number of blocks.

Experimental evaluation on EFG benchmark games is performed using three different dilated regularizers -dilated entropy, dilated global entropy, and dilated $\ell_2$ with different block construction strategies. Experiments demonstrate the benefit of using blocks over non-block based approach in  some games, and no significant diffference between different block construction methods in others.  The results show improved  performance over the state of the art first order method (Mirror-Prox). Further, they introduce a restarting heuristic which speeds up the proposed method as well as the baselines.

**Strengths:**

The paper is generally well written, describing necessary background to help readers understand the paper.

The proposed algorithm seems novel, though I must admit I am not familiar with the literature.

The experimental evaluation seems convincing, the proposed method outperforms SOTA FOM


**Weaknesses:**

line 105-``As discussed before RCMs are not applicable to our setting''. It is not discussed anywhere in the paper why randomized coordinate methods are not applicable.

The paper could include a discussion in the related work about [i]  which develops primal-dual coordinate methods for solving bilinear saddle-point problems.

[i] Carmon et al. Coordinate Methods for Matrix Games

**Questions:**

-

**Limitations:**

The authors discuss open question of why restarting work with regularizers  other than $\ell_2$. The authors adequately addressed the limitations.

---

> ### Author Rebuttal · Authors · 2023-08-10
>
> Thank you for taking the time to review our paper. We provide responses to your concerns below:
>
> > line 105-``As discussed before RCMs are not applicable to our setting''. It is not discussed anywhere in the paper why randomized coordinate methods are not applicable.
>
> We address this in our top-level comment (1, 2). In line 53 of our paper, we mention that in EFGs we do not have the separable structure typically required to use coordinate descent methods. If one is to construct blocks of sequences (which correspond to coordinates in the EFG setting), and sample randomly, there is no guarantee that we can take a gradient step and be at a feasible strategy without projecting back onto the entire sequence-form polytope/treeplex (but this would be too computationally expensive since it would cause the number of (equivalent to) total proximal updates to scale linearly with the number of blocks).  This prevents random sampling of blocks, and requires careful construction and cyclic traversal of the blocks, to ensure that we can generate feasible iterates that will converge ergodically to a Nash equilibrium without incurring a dependence on the number of blocks.  We have made the connection between RCMs not being applicable and lack of separable structure more explicit in a revision of our paper.
>
> > The paper could include a discussion in the related work about [i] which develops primal-dual coordinate methods for solving bilinear saddle-point problems
>
> We appreciate the suggestion and will add this discussion in a revised version of the paper. Briefly: the only setting studied in that paper that is relevant to our work is their $\ell_1-\ell_1$ setting. Their assumption in that case is that the feasible set for each of the players is a probability simplex, which is a much simpler feasible set than the treeplex considered in our work. Importantly, it is unclear how to generalize their result to our setting, as it crucially depends on the simplex structure (see, for example, Eqs. (2) and (5) and the discussion of the data structure design on page 7 in the arXiv version of the cited paper).

---

> > ### Comment · Reviewer_Lrca · 2023-08-15
> >
> > Thank you for your response. I read all the reviews and the rebuttal provided by the authors. I stick to my original rating of "Accept".

---

### Official Review · Reviewer_c8Uw · 2023-07-25

**Soundness:** 2 fair
**Presentation:** 2 fair
**Contribution:** 3 good
**Rating:** 5
**Confidence:** 3

**Summary:**

The proposed Extrapolated Cyclic Primal-Dual Algorithm (ECyclicPDA) is a solution technique for large-scale extensive-form games (EFG) that resembles first-order coordinate descent. To enable pseudo block-wise updates, it takes use of the recursive nature of the proximal update caused by dilated regularizers. A restarting heuristic for EFG solution is also presented by the authors, and it has the potential to significantly speed up both their cyclic technique as well as other current methods like mirror prox and (predictive) CFR+.

**Strengths:**

In summary, the ECyclicPDA algorithm introduces a novel and effective solution technique for tackling large-scale sequential games, demonstrating superior performance compared to current state-of-the-art techniques in some scenarios. The paper examines prior research on coordinate descent methods and first-order techniques for solving EFGs, highlighting the originality and benefits of their proposed ECyclicPDA approach.

**Weaknesses:**

The execution times and computational demands of the ECyclicPDA method in comparison to other cutting-edge approaches are not thoroughly analyzed in the paper. It is essential to evaluate such a comparison in order to ascertain the proposed algorithm's applicability and practical effectiveness. Although it is mentioned that the runtime of ECyclicPDA is independent of the number of blocks, a comprehensive evaluation is still lacking. Additionally, the paper overlooks competitive alternatives like DCFR, despite making references to these methods.

**Questions:**

1) The authors claim, "For the first time, a first-order method has surpassed CFR+ in performance on non-trivial EFGs." However, it seems that in the paper titled "Equivalence Analysis between Counterfactual Regret Minimization and Online Mirror Descent" by Liu et al., the authors also demonstrate instances where first-order OMD methods can outperform CFR+.

2) How strong is the assumption that the strongly convex functions are nice, see page 5, paragraph  "Dilated Regularizers"?

3) In the appendix, the last three figures share the same caption. Do these plots refer to linear, quadratic, and uniform averaging?

**Limitations:**

During the experiments, ECyclicPDA demonstrated superior performance compared to CFR+ solely in the Battleship game. This stands as the only instance where the results were competitive with PCFR+, which still remains the overall best-performing strategy. However, it is not entirely clear from the overall text why ECyclicPDA outperformed CFR+ specifically in the Battleship game.

---

> ### Author Rebuttal · Authors · 2023-08-10
>
> Thank you for taking the time to review our paper. We provide responses to your concerns below:
>
> > The execution times [...] comprehensive evaluation is still lacking.
>
> We address this in our top-level comment (5), the pdf attached to it (containing  numerical evidence that ECyclicPDA scales independently of the number of blocks), and Appendix D.
>
> > Additionally, the paper overlooks competitive alternatives [...]
>
> Predictive CFR$^+$, referred to in our paper as PCFR$^+$, is state-of-the-art among CFR variants for games outside of poker. It is true that specifically in poker-like games, DCFR seems to be better than PCFR$^+$. These conclusions can be found in [14]. We can add DCFR as well if you insist, but it will do worse than regular CFR+ in the non-poker games, and it will not change the takeaways from our experiments and mostly just clutter the plots.
>
> > The authors claim, "For the first time, a first-order method has surpassed CFR+ in performance on non-trivial EFGs." [...]
>
> While we were already aware of that paper, we do thank the reviewer for bringing it to our attention. We would like to argue that the algorithms in Liu et al. are not natural instances of first-order methods, or of OMD/FTRL, for several reasons:
> * The algorithms are what Liu et al. call *future-dependent* OMD/FTRL. “Future-dependent” refers to a very particular design of the regularizer: it has the usual FTRL/OMD regularization of the strategy, but also an additional regularization term that depends on the future strategy. This future strategy has to do with the updates being made in the tree below a given decision point, and is explicitly there to make the algorithm look like CFR/CFR+. Yet, the authors do not show that this odd modification of regularization maintains 1-strong-convexity, a prerequisite for applying OMD/FTRL regret bounds.
> * We do not think it is clear that the proofs in Liu et al are correct. Specifically, their proof of convergence of their future-dependent FTRL/OMD algorithms rely directly on convergence results in [a]. Looking at the proof of Theorem 3.5 in Liu et al, they seem to have at least two major issues:
>   * They invoke Lemma 2 and Theorem 3 of [a] to give a regret bound that they start their derivation from (in equation (59)).  But this is not the bound implied by [a], the bounds implied by [a] are given in equations (A.1) and (A.2) in [a], and they involve several additional terms that must be handled.
>   * Liu et al. state that “Assumptions 1,2,3,5, and 8 in [a] have already been fulfilled.” Assumption 8 in [a] is the assumption that the regularizer is 1-strongly convex with respect to some norm. However, the words “strongly convex” or “strong convexity” do not even appear anywhere in Liu et al., and it is not clear that their time-varying sequence of regularizers satisfies 1-strong-convexity. Our best guess is that it does not, since usually with dilated DGFs the weights need to be chosen carefully in order to secure 1-strong-convexity.
>   * The dilated L2 DGF, which Liu et al. use for their equivalence, is not known to be directionally differentiable, as is required by [a]. Specifically, if an action in an internal decision point is set to zero, then it causes problems with the dilation operation for child decision points. It is unclear what this does to their result. This is a general issue with the dilated L2 DGF, and it causes problems when using it in OMD, because it may require taking the gradient at the relative boundary, which is not well-defined.
>
> We would also like to note that of the three games that overlap between our experiments and theirs, Liar’s Dice, Battleship, and Goofspiel 4 ranks, examining their graphs and ours seems to indicate that we would outperform their algorithms on Battleship and their algorithms would beat us on Goofspiel 4 ranks. We cannot replicate their results for CFR+ for Liar’s Dice (this is evident from the fact that CFR+ reaches a duality gap of $10^{-4}$ before 1000 gradient computations in our experiments, whereas it is not clear if their CFR+ curve reaches an approximate duality gap of $10^{-4}$ even at 4000 gradient computations in their plots). It is not clear why CFR+ would perform so poorly in their experiments.
>
> We are happy to update our paper to discuss the Liu et al. paper. But we prefer not to do so unless the discussion phase for our paper helps us understand whether the Liu et al. paper is correct.
>
> [a] Joulani, Pooria, András György, and Csaba Szepesvári. "A modular analysis of adaptive (non-) convex optimization: Optimism, composite objectives, variance reduction, and variational bounds." Theoretical Computer Science 808 (2020): 108-138.
>
> > How strong is the assumption that the strongly convex functions are nice [...]
>
> The assumption that the strongly convex regularizers are nice is not a strong assumption. This is necessary to develop scalable first-order methods (FOMs) for EFG solving, and is considered in existing FOMs in the EFG literature [13,22,24]. There are several regularizers that are known to be “nice” that lend themselves to use in FOMs for EFG solving. These include the dilated $\ell_2$ regularizer, dilatable global entropy regularizer, and dilated entropy regularizer, which are the three regularizers that we use in our experiments. See [13, 22, 24] which introduce/study these regularizers.
>
> > In the appendix, the last three figures share the same caption [...]
>
> Thank you for pointing this typo out. Yes, figs. 27, 28, 29 depict uniform, linear, and quadratic averaging respectively. We will fix this in a revision.
>
> > During the experiments, ECyclicPDA demonstrated superior performance [...]
>
> We note in our conclusion that we do not know why ECyclicPDA performs better in certain games. Please see the top level comment (6,7) regarding the performance of CFR+. We believe that our introduction of restarting is significant even in the context of PCFR+, given the impressive speedup obtained for some games.

---

### Author Rebuttal · Authors · 2023-08-10

We would like to thank all of the reviewers for taking the time to provide valuable feedback for our paper. Here we provide clarifications for a couple of points raised by multiple reviews. Please note that we have attached a PDF containing tables that we refer to in the rebuttal.

### Theoretical contributions:

(1) The feasible region considered by our problem is non-separable. The feasible region for our problem consists of the strategy space for each of the two players. Players’ strategies are represented using sequences, and the space of these sequences can be formulated as a treeplex, which is essentially a Cartesian product of scaled treeplexes, the base case being that the treeplex is a simplex. (2) If we were to apply a randomized block coordinate method, we would not be able to update one (or a constant number of) those simplices, as (unless we are at a leaf) this would violate feasibility for the entire subtree rooted at the simplex we update. (3) We get around this issue by having a deterministic method that traverses the treeplex bottom-to-top, updating *scaled* values of the coordinates (scaled by the value of the parent node, so each update is on a probability simplex). We can do this thanks to decomposability of the prox update ([12, 22] and stated as Proposition 2.2 in our paper), because we are moving bottom to top, and because another top to bottom traversal fixes the scaling. (4) On a theoretical front, it is surprising that the result we get for such a deterministic, cyclic block coordinate method is independent of the number of blocks (which can be order-dimension) and, to our knowledge, is the first result of this kind for cyclic block coordinate methods. It is also the first block coordinate method whatsoever for EFGs. This would be interesting even if no practical improvements were shown in our experiments.

### Per-iteration complexity of ECyclicPDA:

(5) We consistently beat MP, both in terms of oracle queries and in terms of wall-clock time (attached pdf). We compare to MP because it is a full vector update from the same class of (first-order) methods. We expect similar results for OMD and will include added comparison in a revised version. Several reviewers brought up the per-iteration complexity of our method, with an incorrect understanding that it is more expensive than e.g., the per-iteration cost of MP. The per-iteration complexity of our method is discussed in Appendix D (as is stated in Section 3). In Appendix D, we provide an implementation specific version of our algorithm and a computational complexity analysis of it. We demonstrate that the necessary computations for our method are comparable to those necessary for a full gradient method (i.e., MP). In particular, when first-order methods are applied to solving EFGs, the computational bottleneck are the gradient and proximal update computations (this is noted in lines 38 and 39 in our paper). We will update our paper to emphasize in the main body that the per-iteration complexity is comparable to MP.

In Table 1, we provide a table demonstrating the per-iteration runtime in milliseconds of our algorithm (for each of the different block construction strategies we consider) as well as the algorithms we compare against. It is clear from this table that the runtimes are pretty similar to each other, and this is without extensive optimization of our particular implementation. Clearly, our algorithm is at least as fast as MP per “full” gradient computation. Since the computational bottleneck of gradient and prox computations becomes apparent in bigger games, Battleship demonstrates the speed of our algorithm relative to MP best.

In Table 2, we provide a table demonstrating the wall clock time in seconds of our algorithm to reach a duality gap of $10^{-4}$ (with a timeout of 30 seconds).  It is clear that our algorithm is competitive with MP: MP and its restarted variant time out on Leduc, and MP times out on Battleship (while our algorithm does not even take close to 30 seconds).  Furthermore, we are outperforming CFR+ and its restarted variant in Battleship.

### Restarting as heuristic and comparison to CFR+:

(6) The adaptive restarting heuristic for all methods considered is a contribution of our work since restarting has not been applied to EFG solving previously. We will update the paper to emphasize this by using “r” as a prefix on all algorithms when the restarting heuristic is applied. (7) We would like to note that CFR+ and its variants have a slower convergence rate in the worst case and their good practical performance is still not well understood [11, 12, 23, 24, a]; the strong practical performance of CFR+ and variations is a long-standing open problem in the field and explaining this strong practical performance is outside the scope of the paper. Furthermore, there is recent work [b] which provides evidence that there exist games where the worst-case convergence rates are realized. Nonetheless, we reiterate that our paper is the first paper to *ever* give a first-order method that outperforms CFR+ on EFGs beyond Kuhn poker (which is almost a normal-form game). Secondly, we would like to emphasize that in Fig. 2, the “PCFR+” algorithm is a *new* algorithm, as it is restarted, and restarted PCFR+ had never been considered before; as stated above, we will update the plots to say “rPCFR+” in order to emphasize that it is different from regular PCFR+ (and will update the other restarted versions of other algorithms similarly). Finally, we view our most important contribution as being theoretical in showing the possibility of block-coordinate approaches, as explained above.

[a] Neil Burch. Time and Space: Why Imperfect Information Games are Hard. PhD thesis, University of Alberta, 2017.

[b] Gabriele Farina, Julien Grand-Clément, Christian Kroer, Chung-Wei Lee, and Haipeng Luo. Regret Matching+: (In)Stability and Fast Convergence in Games. arXiv preprint arXiv: 2305.14709, 2023.

---

### Decision · Program_Chairs · 2023-09-21

**Decision:**

Accept (poster)

**Comment:**

We had an especially hard time trying to get reviewers for this paper. Reading the reviews and responses to the reviewers, the contributions to cyclic coordinate descent methods for EFG when the space of strategies for the players is non-separable warrant acceptance.